# Reverse Engineering of Imperceptible Adversarial Image Perturbations

**Yifan Gong**[1][*] **Yuguang Yao**[2][*], **Yize Li**[1], **Yimeng Zhang**[2], **Xiaoming Liu**[2], **Xue Lin**[1], **Sijia Liu**[2,3]
[1] Northeastern University, [2] Michigan State University, [3] MIT-IBM Watson AI Lab, IBM Research
{gong.yifa, li.yize, xue.lin}@northeastern.edu,
{yaoyugua, zhan1853, liusiji5}@msu.edu, liuxm@cse.msu.edu

## Abstract

It has been well recognized that neural network based image classifiers are easily fooled by images with tiny perturbations crafted by an adversary. There has been a vast volume of research to generate and defend such adversarial attacks. However, the following problem is left unexplored: *How to reverse-engineer adversarial perturbations from an adversarial image?* This leads to a new adversarial learning paradigm—Reverse Engineering of Deceptions (RED). If successful, RED allows us to estimate adversarial perturbations and recover the original images. However, carefully crafted, tiny adversarial perturbations are difficult to recover by optimizing a unilateral RED objective. For example, the pure image denoising method may overfit to minimizing the reconstruction error but hardly preserves the classification properties of the true adversarial perturbations. To tackle this challenge, we formalize the RED problem and identify a set of principles crucial to the RED approach design. Particularly, we find that prediction alignment and proper data augmentation (in terms of spatial transformations) are two criteria to achieve a generalizable RED approach. By integrating these RED principles with image denoising, we propose a new Class-Discriminative Denoising based RED framework, termed CDD-RED. Extensive experiments demonstrate the effectiveness of CDD-RED under different evaluation metrics (ranging from the pixel-level, prediction-level to the attribution-level alignment) and a variety of attack generation methods (*e.g.*, FGSM, PGD, CW, AutoAttack, and adaptive attacks). Codes are available at `link`.

## 1 Introduction

Deep neural networks (DNNs) are susceptible to adversarially-crafted tiny input perturbations during inference. Such imperceptible perturbations, *a.k.a.* adversarial attacks, could cause DNNs to draw manifestly wrong conclusions. The existence of adversarial attacks was first uncovered in the domain of image classification (Goodfellow et al., 2014; Carlini & Wagner, 2017; Papernot et al., 2016b), and was then rapidly extended to the other domains, such as object detection (Xie et al., 2017; Serban et al., 2020), language modeling (Cheng et al., 2020; Srikant et al., 2021), and medical machine learning (Finlayson et al., 2019; Antun et al., 2020). Despite different applications, the underlying attack formulations and generation methods commonly obey the ones used in image classification.

A vast volume of existing works have been devoted to designing defenses against such attacks, mostly focusing on either detecting adversarial examples (Grosse et al., 2017; Yang et al., 2020; Metzen et al., 2017; Meng & Chen, 2017; Wójcik et al., 2020) or acquiring adversarially robust DNNs (Madry et al., 2017; Zhang et al., 2019; Wong & Kolter, 2017; Salman et al., 2020; Wong et al., 2020; Carmon et al., 2019; Shafahi et al., 2019). Despite the plethora of prior work on adversarial defenses, it seems impossible to achieve 'perfect' robustness. Given the fact that adversarial attacks are inevitable (Shafahi et al., 2020), we ask whether or not an adversarial attack can be reverse-engineered so that one can estimate the adversary's information (*e.g.*, adversarial perturbations) behind the attack instances. The above problem is referred to as *Reverse Engineering of Deceptions (RED)*, fostering a new adversarial learning regime. The development of RED technologies will also enable the adversarial situation awareness in high-stake applications.

---

[*]Equal contributions.

To the best of our knowledge, few work studied the RED problem. The most relevant one that we are aware of is (Pang et al., 2020), which proposed the so-called query of interest (QOI) estimation model to infer the adversary's target class by model queries. However, the work (Pang et al., 2020) was restricted to the black-box attack scenario and thus lacks a general formulation of RED. Furthermore, it has not built a complete RED pipeline, which should not only provide a solution to estimating the adversarial example but also formalizing evaluation metrics to comprehensively measure the performance of RED. In this paper, we aim to take a solid step towards addressing the RED problem.

## 1.1 CONTRIBUTIONS

The main contributions of our work is listed below.

• We formulate the Reverse Engineering of Deceptions (RED) problem that is able to estimate adversarial perturbations and provides the feasibility of inferring the intention of an adversary, *e.g.*, 'adversary saliency regions' of an adversarial image.

• We identify a series of RED principles to effectively estimate the adversarially-crafted tiny perturbations. We find that the class-discriminative ability is crucial to evaluate the RED performance. We also find that data augmentation, *e.g.*, spatial transformations, is another key to improve the RED result. Furthermore, we integrate the developed RED principles into image denoising and propose a denoiser-assisted RED approach.

• We build a comprehensive evaluation pipeline to quantify the RED performance from different perspectives, such as pixel-level reconstruction error, prediction-level alignment, and attribution-level adversary saliency region recovery. With an extensive experimental study, we show that, compared to image denoising baselines, our proposal yields a consistent improvement across diverse RED evaluation metrics and attack generation methods, *e.g.*, FGSM (Goodfellow et al., 2014), CW (Carlini & Wagner, 2017), PGD (Madry et al., 2017) and AutoAttack (Croce & Hein, 2020).

## 1.2 RELATED WORK

**Adversarial attacks.**    Different types of adversarial attacks have been proposed, ranging from digital attacks (Goodfellow et al., 2014; Carlini & Wagner, 2017; Madry et al., 2017; Croce & Hein, 2020; Xu et al., 2019b; Chen et al., 2017a; Xiao et al., 2018) to physical attacks (Eykholt et al., 2018; Li et al., 2019; Athalye et al., 2018; Chen et al., 2018; Xu et al., 2019c). The former gives the most fundamental threat model that commonly deceives DNN models during inference by crafting imperceptible adversarial perturbations. The latter extends the former to fool the victim models in the physical environment. Compared to digital attacks, physical attacks require much larger perturbation strengths to enhance the adversary's resilience to various physical conditions such as lightness and object deformation (Athalye et al., 2018; Xu et al., 2019c).

In this paper, we focus on $\ell_p$-norm ball constrained attacks, *a.k.a.* $\ell_p$ attacks, for $p \in \{1, 2, \infty\}$, most widely-used in digital attacks. Examples include FGSM (Goodfellow et al., 2014), PGD (Madry et al., 2017), CW (Carlini & Wagner, 2017), and the recently-released attack benchmark AutoAttack (Croce & Hein, 2020). Based on the adversary's intent, $\ell_p$ attacks are further divided into untargeted attacks and targeted attacks, where in contrast to the former, the latter designates the (incorrect) prediction label of a victim model. When an adversary has no access to victim models' detailed information (such as architectures and model weights), $\ell_p$ attacks can be further generalized to black-box attacks by leveraging either surrogate victim models (Papernot et al., 2017; 2016a; Dong et al., 2019; Liu et al., 2017) or input-output queries from the original black-box models (Chen et al., 2017b; Liu et al., 2019; Cheng et al., 2019).

**Adversarial defenses.**    To improve the robustness of DNNs, a variety of approaches have been proposed to defend against $\ell_p$ attacks. One line of research focuses on enhancing the robustness of DNNs during training, *e.g.*, adversarial training (Madry et al., 2017), TRADES (Zhang et al., 2019), randomized smoothing (Wong & Kolter, 2017), and their variants (Salman et al., 2020; Wong et al., 2020; Carmon et al., 2019; Shafahi et al., 2019; Uesato et al., 2019; Chen et al., 2020). Another line of research is to detect adversarial attacks without altering the victim model or the training process. The key technique is to differentiate between benign and adversarial examples by measuring their 'distance.' Such a distance measure has been defined in the input space via pixel-level reconstruction error (Meng & Chen, 2017; Liao et al., 2018), in the intermediate layers via neuron

activation anomalies (Xu et al., 2019a), and in the logit space by tracking the sensitivity of deep feature attributions to input perturbations (Yang et al., 2020).

In contrast to RED, *adversarial detection is a relatively simpler problem* as a roughly approximated distance possesses detection-ability (Meng & Chen, 2017; Luo et al., 2015). Among the existing adversarial defense techniques, the recently-proposed Denoised Smoothing (DS) method (Salman et al., 2020) is more related to ours. In (Salman et al., 2020), an image denoising network is prepended to an existing victim model so that the augmented system can be performed as a smoothed image classifier with certified robustness. Although DS is not designed for RED, its denoised output can be regarded as a benign example estimate. The promotion of classification stability in DS also motivates us to design the RED methods with class-discriminative ability. Thus, DS will be a main baseline approach for comparison. Similar to our RED setting, the concurrent work (Souri et al., 2021) also identified the feasibility of estimating adversarial perturbations from adversarial examples.

## 2 REVERSE ENGINEERING OF DECEPTIONS: FORMULATION AND CHALLENGES

In this section, we first introduce the threat model of our interest: adversarial attacks on images. Based on that, we formalize the Reverse Engineering of Deceptions (RED) problem and demonstrate its challenges through some 'warm-up' examples.

**Preliminaries on threat model.** We focus on $\ell_p$ attacks, where the *adversary's goal* is to generate imperceptible input perturbations to fool a well-trained image classifier. Formally, let $\mathbf{x}$ denote a benign image, and $\boldsymbol{\delta}$ an additive perturbation variable. Given a victim classifier $f$ and a perturbation strength tolerance $\epsilon$ (in terms of, *e.g.*, $\ell_\infty$-norm constraint $\|\boldsymbol{\delta}\|_\infty \le \epsilon$), the desired *attack generation algorithm* $\mathcal{A}$ then seeks the optimal $\boldsymbol{\delta}$ subject to the perturbation constraints. Such an attack generation process is denoted by $\boldsymbol{\delta} = \mathcal{A}(\mathbf{x}, f, \epsilon)$, resulting in an adversarial example $\mathbf{x}' = \mathbf{x} + \boldsymbol{\delta}$. Here $\mathcal{A}$ can be fulfilled by different attack methods, e.g., FGSM (Goodfellow et al., 2014), CW (Carlini & Wagner, 2017), PGD (Madry et al., 2017), and AutoAttack (Croce & Hein, 2020).

**Problem formulation of RED.** Different from conventional defenses to detect or reject adversarial instances (Pang et al., 2020; Liao et al., 2018; Shafahi et al., 2020; Niu et al., 2020), RED aims to address the following question.

> **(RED problem)** Given an adversarial instance, can we reverse-engineer the adversarial perturbations $\boldsymbol{\delta}$, and infer the adversary's objective and knowledge, *e.g.*, true image class behind deception and adversary saliency image region?

Formally, we aim to recover $\boldsymbol{\delta}$ from an adversarial example $\mathbf{x}'$ under the prior knowledge of the victim model $f$ or its substitute $\hat{f}$ if the former is a black box. We denote the RED operation as $\boldsymbol{\delta} = \mathcal{R}(\mathbf{x}', \hat{f})$, which covers the white-box scenario ($\hat{f} = f$) as a special case. We propose to learn a parametric model $\mathcal{D}_{\boldsymbol{\theta}}$ (*e.g.*, a denoising neural network that we will focus on) as an approximation of $\mathcal{R}$ through a training dataset of adversary-benignity pairs $\Omega = \{(\mathbf{x}', \mathbf{x})\}$. Through $\mathcal{D}_{\boldsymbol{\theta}}$, RED will provide a **benign example estimate** $\mathbf{x}_{\mathrm{RED}}$ and a **adversarial example estimate** $\mathbf{x}'_{\mathrm{RED}}$ as below:

$$\mathbf{x}_{\mathrm{RED}} = \mathcal{D}_{\boldsymbol{\theta}}(\mathbf{x}'), \quad \mathbf{x}'_{\mathrm{RED}} = \underbrace{\mathbf{x}' - \mathbf{x}_{\mathrm{RED}}}_{\text{perturbation estimate}} + \mathbf{x}, \tag{1}$$

where a **perturbation estimate** is given by subtracting the RED's output with its input, $\mathbf{x}' - \mathcal{D}_{\boldsymbol{\theta}}(\mathbf{x}')$.

We **highlight** that RED yields a new defensive approach aiming to 'diagnose' the perturbation details of an existing adversarial example in a post-hoc, forensic manner. This is different from adversarial detection (AD). Fig.1 provides a visual comparison of RED with AD. Although AD is also designed in a post-hoc manner, it aims to determine whether an input is an adversarial example for a victim model based on certain statistics on model features or logits. Besides, AD might be used as a pre-processing step of RED, where the former provides 'detected' adversarial examples for fine-level RED diagnosis. In our experiments, we will

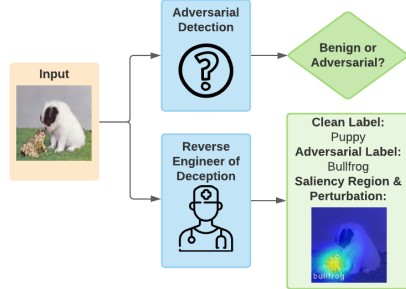

Figure 1: Overview of RED versus AD.

also show that the outputs of RED can be leveraged to guide the design of adversarial detection. In this sense, RED and AD are complementary building blocks within a closed loop.

**Challenges of RED**    In this work, we will specify the RED model $\mathcal{D}_{\boldsymbol{\theta}}$ as a denoising network. However, it is highly non-trivial to design a proper denoiser for RED. Speaking at a high level, there exist two main challenges. First, unlike the conventional image denoising strategies (Zhang et al., 2017), the design of an RED-aware denoiser needs to take into account the effects of victim models and data properties of adversary-benignity pairs. Second, it might be insufficient to merely minimize the reconstruction error as the adversarial perturbation is finely-crafted (Niu et al., 2020). Therefore, either under- or over-denoising will lead to poor RED performance.

## 3    RED EVALUATION METRICS AND DENOISING-ONLY BASELINE

Since RED is different from existing defensive approaches, we first develop new performance metrics of RED, ranging from pixel-level reconstruction error to attribution-level adversary saliency region. We next leverage the proposed performance metrics to demonstrate why a pure image denoiser is incapable of fulfilling RED.

**RED evaluation metrics.**    Given a learned RED model $\mathcal{D}_{\boldsymbol{\theta}}$, the RED performance will be evaluated over a testing dataset $(\mathbf{x}', \mathbf{x}) \in \mathcal{D}_{\text{test}}$; see implementation details in Sec. 5. Here, $\mathbf{x}'$ is used as the testing input of the RED model, and $\mathbf{x}$ is the associated ground-truth benign example for comparison. The benign example estimate $\mathbf{x}_{\text{RED}}$ and adversarial example estimate $\mathbf{x}'_{\text{RED}}$ are obtained following (1). RED evaluation pipeline is conducted from the following aspects: ① pixel-level reconstruction error, ② prediction-level inference alignment, and ③ attribution-level adversary saliency region.

➢ ① **Pixel-level**: Reconstruction error given by $d(\mathbf{x}, \mathbf{x}_{\text{RED}}) = \mathbb{E}_{(\mathbf{x}', \mathbf{x}) \in \mathcal{D}_{\text{test}}} [\|\mathbf{x}_{\text{RED}} - \mathbf{x}\|_2]$.

➢ ② **Prediction-level**: Prediction alignment (PA) between the pair of *benign* example and its estimate $(\mathbf{x}_{\text{RED}}, \mathbf{x})$ and PA between the pair of *adversarial* example and its estimate $(\mathbf{x}'_{\text{RED}}, \mathbf{x}')$, given by

$$\text{PA}_{\text{benign}} = \frac{\text{card}(\{(\mathbf{x}_{\text{RED}}, \mathbf{x}) \,|\, F(\mathbf{x}_{\text{RED}}) = F(\mathbf{x})\})}{\text{card}(\mathcal{D}_{\text{test}})}, \; \text{PA}_{\text{adv}} = \frac{\text{card}(\{(\mathbf{x}'_{\text{RED}}, \mathbf{x}') \,|\, F(\mathbf{x}'_{\text{RED}}) = F(\mathbf{x}')\})}{\text{card}(\mathcal{D}_{\text{test}})}$$

where $\text{card}(\cdot)$ denotes a cardinality function of a set and $F$ refers to the prediction label provided by the victim model $f$.

➢ ③ **Attribution-level**: Input attribution alignment (IAA) between the benign pair $(\mathbf{x}_{\text{RED}}, \mathbf{x})$ and between the adversarial pair $(\mathbf{x}'_{\text{RED}}, \mathbf{x}')$. In this work, we adopt GradCAM (Selvaraju et al., 2020) to attribute the predictions of classes back to input saliency regions. The rationale behind IAA is that the unnoticeable adversarial perturbations (in the pixel space) can introduce an evident input attribution discrepancy with respect to (w.r.t.) the true label $y$ and the adversary's target label $y'$ (Boopathy et al., 2020; Xu et al., 2019b). Thus, an accurate RED should be able to erase the adversarial attribution effect through $\mathbf{x}_{\text{RED}}$, and estimate the adversarial intent through the saliency region of $\mathbf{x}'_{\text{RED}}$ (see Fig. 1 for illustration).

**Denoising-Only (DO) baseline.**    We further show that how a pure image denoiser, a 'must-try' baseline, is insufficient of tackling the RED problem. This failure case drive us to rethink the denoising strategy through the lens of RED. First, we obtain the denoising network by minimizing the reconstruction error:

Figure 2:  IAA of DO compared with ground-truth.

$$\underset{\boldsymbol{\theta}}{\text{minimize}} \quad \ell_{\text{denoise}}(\boldsymbol{\theta}; \Omega) := \mathbb{E}_{(\mathbf{x}', \mathbf{x}) \in \Omega} \|\mathcal{D}_{\boldsymbol{\theta}}(\mathbf{x}') - \mathbf{x}\|_1, \tag{2}$$

where a Mean Absolute Error (MAE)-type loss is used for denoising (Liao et al., 2018), and the creation of training dataset $\Omega$ is illustrated in Sec. 5.1. Let us then evaluate the performance of DO through the non-adversarial prediction alignment $\text{PA}_{\text{benign}}$ and IAA. We find that $\text{PA}_{\text{benign}} = 42.8\%$ for DO. And Fig. 2 shows the IAA performance of DO w.r.t. an input example. As we can see, DO is not capable of exactly recovering the adversarial saliency regions compared to the ground-truth

adversarial perturbations. These suggest that DO-based RED lacks the reconstruction ability at the prediction and the attribution levels. Another naive approach is performing adversarial attack back to $x'$, yet it requires additional assumptions and might not precisely recover the ground-truth perturbations. The detailed limitations are discussed in Appendix A.

## 4 CLASS-DISCRIMINATIVE DENOISING FOR RED

In this section, we propose a novel Class-Discriminative Denoising based RED approach termed CDD-RED; see Fig. 3 for an overview. CDD-RED contains two key components. First, we propose a PA regularization to enforce the prediction-level stabilities of both estimated benign example $\mathbf{x}_{\mathrm{RED}}$ and adversarial example $\mathbf{x}'_{\mathrm{RED}}$ with respect to their true counterparts $\mathbf{x}$ and $\mathbf{x}'$, respectively. Second, we propose a data augmentation strategy to improve the RED's generalization without losing its class-discriminative ability.

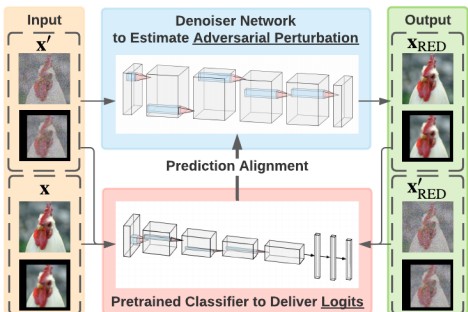

Figure 3: CDD-RED overview.

**Benign and adversarial prediction alignment.** To accurately estimate the adversarial perturbation from an adversarial instance, the lessons from the DO approach suggest to preserve the class-discriminative ability of RED estimates to align with the original predictions, given by $\mathbf{x}_{\mathrm{RED}}$ vs. $\mathbf{x}$, and $\mathbf{x}'_{\mathrm{RED}}$ vs. $\mathbf{x}'$. Spurred by that, the training objective of CDD-RED is required not only to minimize the reconstruction error like (2) but also to maximize PA, namely, 'clone' the class-discriminative ability of original data. To achieve this goal, we augment the denoiser $\mathcal{D}_{\boldsymbol{\theta}}$ with a known classifier $\hat{f}$ to generate predictions of estimated benign and adversarial examples (see Fig. 3), *i.e.*, $\mathbf{x}_{\mathrm{RED}}$ and $\mathbf{x}'_{\mathrm{RED}}$ defined in (1). By contrasting $\hat{f}(\mathbf{x}_{\mathrm{RED}})$ with $\hat{f}(\mathbf{x})$, and $\hat{f}(\mathbf{x}'_{\mathrm{RED}})$ with $\hat{f}(\mathbf{x}')$, we can promote PA by minimizing the prediction gap between true examples and estimated ones:

$$\ell_{\mathrm{PA}}(\boldsymbol{\theta};\Omega) = \mathbb{E}_{(\mathbf{x}',\mathbf{x})\in\Omega}[\ell_{\mathrm{PA}}(\boldsymbol{\theta};\mathbf{x}',\mathbf{x})], \ \ell_{\mathrm{PA}}(\boldsymbol{\theta};\mathbf{x}',\mathbf{x}) := \underbrace{\mathrm{CE}(\hat{f}(\mathbf{x}_{\mathrm{RED}}),\hat{f}(\mathbf{x}))}_{\text{PA for benign prediction}} + \underbrace{\mathrm{CE}(\hat{f}(\mathbf{x}'_{\mathrm{RED}}),\hat{f}(\mathbf{x}'))}_{\text{PA for adversarial prediction}}, \quad (3)$$

where CE denotes the cross-entropy loss. To enhance the class-discriminative ability, it is desirable to integrate the denoising loss (2) with the PA regularization (3), leading to $\ell_{\mathrm{denoise}} + \lambda\ell_{\mathrm{PA}}$, where $\lambda > 0$ is a regularization parameter. To address this issue, we will further propose a data augmentation method to improve the denoising ability without losing the advantage of PA regularization.

**Proper data augmentation improves RED.** The rationale behind incorporating image transformations into CDD-RED lies in two aspects. First, data transformation can make RED foveated to the most informative attack artifacts since an adversarial instance could be sensitive to input transformations (Luo et al., 2015; Athalye et al., 2018; Xie et al., 2019; Li et al., 2020; Fan et al., 2021). Second, the identification of transformation-resilient benign/adversarial instances may enhance the capabilities of PA and IAA.

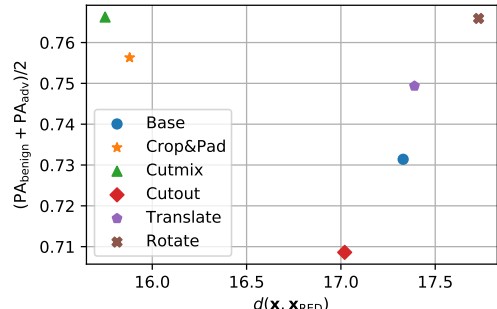

Figure 4: The influence of different data augmentations. 'Base' refers to the base training without augmentation.

However, it is highly non-trivial to determine the most appropriate data augmentation operations. For example, a pixel-sensitive data transformation, e.g., Gaussian blurring and colorization, would hamper the reconstruction-ability of the original adversary-benignity pair $(\mathbf{x}', \mathbf{x})$. Therefore, we focus on spatial image transformations, including rotation, translation, cropping & padding, cutout, and CutMix (Yun et al., 2019), which keep the original perturbation in a linear way. In Fig.4, we evaluate the RED performance, in terms of pixel-level reconstruction error and prediction-level alignment accuracy, for different kinds of spatial image

transformations. As we can see, CutMix and cropping & padding can increase the both performance simultaneously, considered as the appropriate augmentation to boost the RED. Furthermore, we empirically find that combining the two transformations can further improve the performance.

Let $\mathcal{T}$ denote a transformation set, including cropping & padding and CutMix operations. With the aid of the denoising loss (2), PA regularization (3), and data transformations $\mathcal{T}$, we then cast the overall training objective of CDD-RED as

$$\underset{\boldsymbol{\theta}}{\text{minimize}} \quad \underbrace{\mathbb{E}_{(\mathbf{x}',\mathbf{x})\in\Omega,t\sim\mathcal{T}}\|\mathcal{D}_{\boldsymbol{\theta}}(t(\mathbf{x}')) - t(\mathbf{x})\|_1}_{\ell_{\text{denoise}}\ (2)\ \text{with data augmentations}} + \underbrace{\lambda\mathbb{E}_{(\mathbf{x}',\mathbf{x})\in\Omega,t\sim\check{\mathcal{T}}}[\ell_{\text{PA}}(\boldsymbol{\theta};t(\mathbf{x}'),t(\mathbf{x}))]}_{\ell_{\text{PA}}\ (3)\ \text{with data augmentation via}\ \check{\mathcal{T}}}, \quad (4)$$

where $\check{\mathcal{T}}$ denotes a properly-selected subset of $\mathcal{T}$, and $\lambda > 0$ is a regularization parameter. In the PA regularizer (4), we need to avoid the scenario of over-transformation where data augmentation alters the classifier's original decision. This suggests $\check{\mathcal{T}} = \{t \in \mathcal{T} \mid \hat{F}(t(\mathbf{x})) = \hat{F}(\mathbf{x}), \hat{F}(t(\mathbf{x}')) = \hat{F}(\mathbf{x}')\}$, where $\hat{F}$ represents the prediction label of the pre-trained classifier $\hat{f}$, *i.e.*, $\hat{F}(\cdot) = \text{argmax}(\hat{f}(\cdot))$.

## 5 EXPERIMENTS

We show the effectiveness of our proposed method in 5 aspects: **a)** reconstruction error of adversarial perturbation inversion, *i.e.*, $d(\mathbf{x}, \mathbf{x}_{\text{RED}})$, **b)** class-discriminative ability of the benign and adversarial example estimate, *i.e.*, $\text{PA}_{\text{benign}}$ and $\text{PA}_{\text{adv}}$ by victim models, **c)** adversary saliency region recovery, *i.e.*, attribution alignment, and **d)** RED evaluation over unseen attack types and adaptive attacks.

### 5.1 EXPERIMENT SETUP

**Attack datasets.** To train and test RED models, we generate adversarial examples on the ImageNet dataset (Deng et al., 2009). We consider 3 **attack methods** including PGD (Madry et al., 2017), FGSM (Goodfellow et al., 2014), and CW attack (Carlini & Wagner, 2017), applied to 5 **models** including pre-trained ResNet18 (Res18), Resnet50 (Res50) (He et al., 2015), VGG16, VGG19, and InceptionV3 (IncV3) (Szegedy et al., 2015). The detailed parameter settings can be found in Appendix B. Furthermore, to evaluate the RED performance on unseen perturbation types during training, additional 2K adversarial examples generated by **AutoAttack** (Croce & Hein, 2020) and 1K adversarial examples generated by **Feature Attack** (Sabour et al., 2015) are included as the unseen testing dataset. AutoAttack is applied on VGG19, Res50 and **two new victim models**, i.e., Alexet and Robust Resnet50 (R-Res50), via fast adversarial training (Wong et al., 2020) while Feature Attack is applied on VGG19 and Alexnet. The rational behind considering Feature Attack is that feature adversary has been recognized as an effective way to circumvent adversarial detection (Tramer et al., 2020). Thus, it provides a supplement on detection-aware attacks.

**RED model configuration, training and evaluation.** During the training of the RED denoisers, VGG19 (Simonyan & Zisserman, 2015) is chosen as the pretrained classifier $\hat{f}$ for PA regularization. Although different victim models were used for generating adversarial examples, we will show that the inference guided by VGG19 is able to accurately estimate the true image class and the intent of the adversary. In terms of the architecture of $\mathcal{D}_{\boldsymbol{\theta}}$, DnCNN (Zhang et al., 2017) is adopted. The RED problem is solved using an Adam optimizer (Kingma & Ba, 2015) with the initial learning rate of $10^{-4}$, which decays 10 times for every 140 training epochs. In (4), the regularization parameter $\lambda$ is set as 0.025. The transformations for data augmentation include CutMix and cropping & padding. The maximum number of training epochs is set as 300. The computation cost and ablation study of CDD-RED are in Appendix D and E, respectively.

**Baselines.** We compare CDD-RED with two baseline approaches: **a)** the conventional denoising-only (DO) approach with the objective function (2); **b)** The state-of-the-art Denoised Smoothing (DS) (Salman et al., 2020) approach that considers both the reconstruction error and the PA for benign examples in the objective function. Both methods are tuned to their best configurations.

### 5.2 MAIN RESULTS

**Reconstruction error $d(\mathbf{x}, \mathbf{x}_{\text{RED}})$ and PA.** Table 1 presents the comparison of CDD-RED with the baseline denoising approaches in terms of $d(\mathbf{x}, \mathbf{x}_{\text{RED}})$, $d(f(\mathbf{x}), f(\mathbf{x}_{\text{RED}}))$, $d(f(\mathbf{x}'), f(\mathbf{x}'_{\text{RED}}))$,

$PA_{benign}$, and $PA_{adv}$ on the testing dataset. As we can see, our approach (CDD-RED) improves the class-discriminative ability from benign perspective by 42.91% and adversarial perspective by 8.46% with a slightly larger reconstruction error compared with the DO approach. In contrast to DS, CDD-RED achieves similar $PA_{benign}$ but improved pixel-level denoising error and $PA_{adv}$. Furthermore, CDD-RED achieves the best logit-level reconstruction error for both $f(\mathbf{x}_{RED})$ and $f(\mathbf{x}'_{RED})$ among the three approaches. This implies that $\mathbf{x}_{RED}$ rendered by CDD-RED can achieve highly similar prediction to the true benign example $\mathbf{x}$, and the perturbation estimate $\mathbf{x}' - \mathbf{x}_{RED}$ yields a similar misclassification effect to the ground-truth perturbation. Besides, CDD-RED is robust against attacks with different hyperparameters settings, details can be found in Appendix F.

Table 1: The performance comparison among DO, DS and CDD-RED on the testing dataset.

|  | DO | DS | CDD-RED |
|---|---|---|---|
| $d(\mathbf{x}, \mathbf{x}_{RED})$ | 9.32 | 19.19 | 13.04 |
| $d(f(\mathbf{x}), f(\mathbf{x}_{RED}))$ | 47.81 | 37.21 | 37.07 |
| $d(f(\mathbf{x}'), f(\mathbf{x}'_{RED}))$ | 115.09 | 150.02 | 78.21 |
| $PA_{benign}$ | 42.80% | 86.64% | 85.71% |
| $PA_{adv}$ | 71.97% | 72.47% | 80.43% |

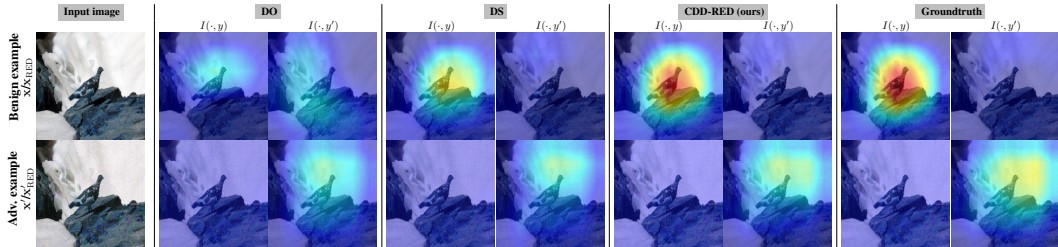

Figure 5: Interpretation ($I$) of benign ($\mathbf{x}/\mathbf{x}_{RED}$) and adversarial ($\mathbf{x}'/\mathbf{x}'_{RED}$) image w.r.t. the true label $y$='ptarmigan' and the adversary targeted label $y'$='shower curtain'. We compare three methods of RED training, DO, DS, and CDD-RED as our method, to the ground-truth interpretation. Given an RED method, the first column is $I(\mathbf{x}_{RED}, y)$ versus $I(\mathbf{x}'_{RED}, y)$, the second column is $I(\mathbf{x}_{RED}, y')$ versus $I(\mathbf{x}'_{RED}, y')$, and all maps under each RED method are normalized w.r.t. their largest value. For the ground-truth, the first column is $I(\mathbf{x}, y)$ versus $I(\mathbf{x}', y)$, the second column is $I(\mathbf{x}, y')$ versus $I(\mathbf{x}', y')$.

**Attribution alignment.** In addition to pixel-level alignment and prediction-level alignment to evaluate the RED performance, attribution alignment is examined in what follows. Fig. 5 presents attribution maps generated by GradCAM in terms of $I(\mathbf{x}, y)$, $I(\mathbf{x}', y)$, $I(\mathbf{x}, y')$, and $I(\mathbf{x}', y')$, where $\mathbf{x}'$ denotes the perturbed version of $\mathbf{x}$, and $y'$ is the adversarially targeted label. From left to right is the attribution map over DO, DS, CDD-RED (our method), and the ground-truth. Compared with DO and DS, CDD-RED yields a closer attribution alignment with the ground-truth especially when making a comparison between $I(\mathbf{x}_{RED}, y)$ and $I(\mathbf{x}, y)$. At the dataset level, Fig. 6 shows the distribution of attribution IoU scores. It is observed that the IoU distribution of CDD-RED, compared with DO and DS, has a denser concentration over the high-value area, corresponding to closer alignment with the attribution map by the adversary. This feature indicates an interesting application of the proposed RED approach, which is to achieve the recovery of adversary's saliency region, in terms of the class-discriminative image regions that the adversary focused on.

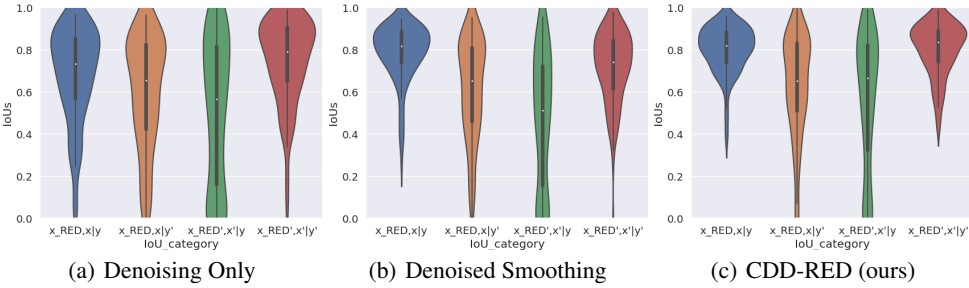

|  |  |  |
|---|---|---|
| (a) Denoising Only | (b) Denoised Smoothing | (c) CDD-RED (ours) |

Figure 6: IoU distributions of the attribution alignment by three RED methods. Higher IoU is better. For each subfigure, the four IoU scores standing for $IoU(\mathbf{x}_{RED}, \mathbf{x}, y)$, $IoU(\mathbf{x}_{RED}, \mathbf{x}, y')$, $IoU(\mathbf{x}'_{RED}, \mathbf{x}', y)$, and $IoU(\mathbf{x}'_{RED}, \mathbf{x}', y')$.

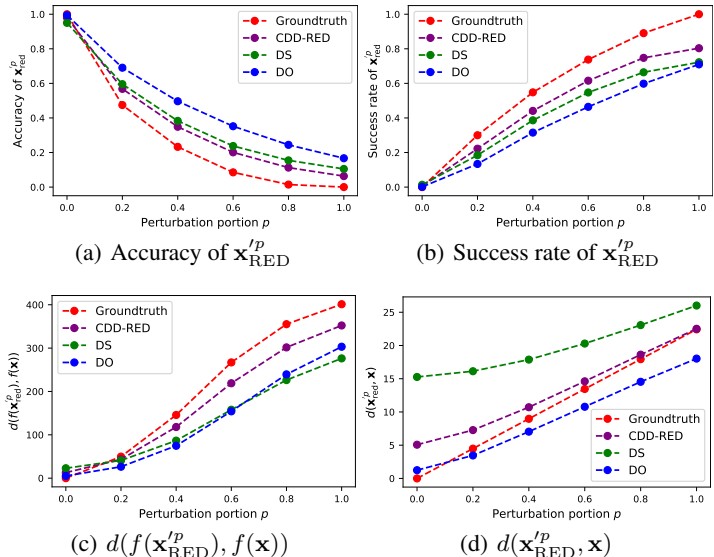

(a) Accuracy of $\mathbf{x}'^p_{\mathrm{RED}}$

(b) Success rate of $\mathbf{x}'^p_{\mathrm{RED}}$

(c) $d(f(\mathbf{x}'^p_{\mathrm{RED}}), f(\mathbf{x}))$

(d) $d(\mathbf{x}'^p_{\mathrm{RED}}, \mathbf{x})$

Figure 7: Reverse engineer partially-perturbed data under different interpolation portion $p$.

**RED vs. unforeseen attack types.** The experiments on the recovery of unforeseen attack types are composed of two parts: **a)** partially-perturbed data via linear interpolation, and **b)** the unseen attack type, AutoAttack, Feature Attack, and Adaptive Attack. More attack results including adversarial attacks on CIFAR-10 dataset, Wasserstein minimized attacks, and attacks on smoothed classifiers can be found in Appendix C.

First, we construct partially-perturbed data by adding a portion $p \in \{0\%, 20\%, \cdots, 100\%\}$ of the perturbation $\mathbf{x}' - \mathbf{x}$ to the true benign example $\mathbf{x}$, namely, $\mathbf{x}'^p = \mathbf{x} + p(\mathbf{x}' - \mathbf{x})$. The interpolated $\mathbf{x}'^p$ is then used as the input to an RED model. We aim to investigate whether or not the proposed RED method can recover partial perturbations (even not successful attacks).

Fig. 7 (a) and (b) show the the prediction alignment with $y$ and $y'$, of the adversarial example estimate $\mathbf{x}'^p_{\mathrm{RED}} = \mathbf{x}'^p - \mathcal{D}_{\boldsymbol{\theta}}(\mathbf{x}'^p) + \mathbf{x}$ by different RED models. Fig. 7 (c) shows the logit distance between the prediction of the partially-perturbed adversarial example estimate and the prediction of the benign example while Fig. 7 (d) demonstrates the pixel distance between $\mathbf{x}'^p_{\mathrm{RED}}$ and the benign example.

First, a smaller gap between the ground-truth curve (in red) and the adversarial example estimate $\mathbf{x}'^p_{\mathrm{red}}$ curve indicates a better performance. Fig. 7 (a) and (b) show that CDD-RED estimates the closest adversary's performance to the ground truth in terms of the prediction accuracy and attack success rate. This is also verified by the distance of prediction logits in Fig. 7 (c). Fig. 7 (d) shows that DS largely over-estimates the additive perturbation, while CDD-RED maintains the perturbation estimation performance closest to the ground truth. Though DO is closer to the ground-truth than CDD-RED at p < 40%, DO is not able to recover a more precise adversarial perturbation in terms of other performance metrics. For example, in Fig. 7 (b) at p = 0.2, $\mathbf{x}'^p_{\mathrm{RED}}$ by DO achieves a lower successful attack rate compared to CDD-RED and the ground-truth.

Moreover, as for benign examples with $p = 0\%$ perturbations, though the RED denoiser does not see benign example pair $(\mathbf{x}, \mathbf{x})$ during training, it keeps the performance of the benign example recovery. CDD-RED can handle the case with a mixture of adversarial and benign examples. That is to say, even if a benign example, detected as adversarial, is wrongly fed into the RED framework, our method can recover the original perturbation close to the ground truth. See Appendix G for details.

Table 2: The $d(\mathbf{x}, \mathbf{x}_{\mathrm{RED}})$, $\mathrm{PA}_{\mathrm{benign}}$, and $\mathrm{PA}_{\mathrm{adv}}$ performance of the denoisers on the unforeseen perturbation type AutoAttack, Feature Attack, and Adaptive Attack.

| | | DO | DS | CDD-RED |
|---|---|---|---|---|
| $d(\mathbf{x}, \mathbf{x}_{\mathrm{RED}})$ | AutoAttack | 6.41 | 16.64 | 8.81 |
| | Feature Attack | 5.51 | 16.14 | 7.99 |
| | Adaptive Attack | 9.76 | 16.21 | 12.24 |
| $\mathrm{PA}_{\mathrm{benign}}$ | AutoAttack | 84.69% | 92.64% | 94.58% |
| | Feature Attack | 82.90% | 90.75% | 93.25% |
| | Adaptive Attack | 33.20% | 27.27% | 36.29% |
| $\mathrm{PA}_{\mathrm{adv}}$ | AutoAttack | 85.53% | 83.30% | 88.39% |
| | Feature Attack | 26.97% | 35.84% | 63.48% |
| | Adaptive Attack | 51.21% | 55.41% | 57.11% |

Table 2 shows the RED performance on the unseen attack type, AutoAttack, Feature Attack, and Adaptive Attack. For AutoAttack and Feature Attack, CDD-RED outperforms both DO and DS in terms of PA from both benign and adversarial perspectives. Specifically, CDD-RED increases the $PA_{adv}$ for Feature Attack by 36.51% and 27.64% compared to DO and DS, respectively.

As for the adaptive attack (Tramer et al., 2020), we assume that the attacker has access to the knowledge of the RED model, *i.e.*, $D_\theta$. It can then perform the PGD attack method to generate successful prediction-evasion attacks even after taking the RED operation.

We use PGD methods to generate such attacks within the $\ell_\infty$-ball of perturbation radius $\epsilon = 20/255$. Table 2 shows that Adaptive Attack is much stronger than Feature Attack and AutoAttack, leading to larger reconstruction error and lower PA. However, CDD-RED still outperforms DO and DS in $PA_{benign}$ and $PA_{adv}$. Compared to DS, it achieves a better trade-off with denoising error $d(\mathbf{x}, \mathbf{x}_{RED})$.

In general, CDD-RED can achieve high PA even for unseen attacks, indicating the generalization-ability of our method to estimate not only new adversarial examples (generated from the same attack method), but also new attack types.

**RED to infer correlation between adversaries.** In what follows, we investigate whether the RED model guided by the single classifier (VGG19) enables to identify different adversary classes, given by combinations of attack types (FGSM, PGD, CW) and victim model types (Res18, Res50, VGG16, VGG19, IncV3).

Fig. 8 presents the correlation between every two adversary classes in the logit space. Fig. 8 (a) shows the ground-truth correlation map. Fig. 8 (b) shows correlations between logits of $\mathbf{x}'_{RED}$ estimated by our RED method (CDD-RED) and logits of the true $\mathbf{x}'$. Along the diagonal of each correlation matrix, the darker implies the better RED estimation under the same adversary class.

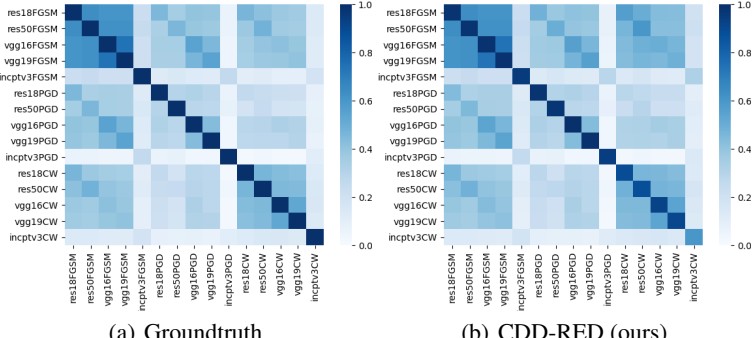

(a) Groundtruth        (b) CDD-RED (ours)

Figure 8: Correlation matrices between different adversaries. For each correlation matrix, rows and columns represent adversarial example estimate $\mathbf{x}'_{RED}$ and true adversarial example $\mathbf{x}'$ (For the ground-truth correlation matrix, $\mathbf{x}'_{RED} = \mathbf{x}'$). Each entry represents the average Spearman rank correlation between the logits of two adversary settings $\in$ {(victim model, attack type)}.

By peering into off-diagonal entries, we find that FGSM attacks are more resilient to the choice of a victim model (see the cluster of high correlation values at the top left corner of Fig. 8). Meanwhile, the proposed CDD-RED precisely recovers the correlation behavior of the true adversaries. Such a correlation matrix can help explain the similarities between different attacks' properties. Given an inventory of existing attack types, if a new attack appears, then one can resort to RED to estimate the correlations between the new attack type and the existing attack types. Based on the correlation screening, it can infer the properties of the new attack type based on its most similar counterpart in the existing attack library; see Appendix I.2. Inspired by the correlation, RED-synthesized perturbations can also be used as a transfer attack as well; see Appendix H.

**Other applications of RED.** In Appendix I, we also empirically show that the proposed RED approach can be applied to adversarial detection, attack identity inference, and adversarial defense.

## 6 CONCLUSION

In this paper, we study the problem of Reverse Engineering of Deceptions (RED), to recover the attack signatures (*e.g.* adversarial perturbations and adversary saliency regions) from an adversarial instance. To the best of our knowledge, RED has not been well studied. Our work makes a solid step towards formalizing the RED problem and developing a systematic pipeline, covering not only a solution but also a complete set of evaluation metrics. We have identified a series of RED principles, ranging from the pixel level to the attribution level, desired to reverse-engineer adversarial attacks. We have developed an effective denoiser-assisted RED approach by integrating class-discrimination and data augmentation into an image denoising network. With extensive experiments, our approach outperforms the existing baseline methods and generalizes well to unseen attack types.

## ACKNOWLEDGMENT

The work is supported by the DARPA RED program. We also thank the MIT-IBM Watson AI Lab, IBM Research for supporting us with GPU computing resources.

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

## A  RED BY PGD ATTACK: PERFORMING PGD BACK TO THE TRUE CLASS

A naive approach to reverse engineer the adversarial perturbation is using the target PGD attack to revert the label back to the groundtruth. However, this requires additional assumptions. First, since PGD is a test-time deterministic optimization approach for perturbation generation, its targeted implementation requires the true class of the adversarial example, which could be unknown at testing time. What is more, one has to pre-define the perturbation budget $\epsilon$ for PGD. This value is also unknown. Second, performing PGD back to the true class might not exactly recover the ground-truth adversarial perturbations. By contrast, its RED counterpart could be over-perturbed. To make it more convincing, we applied the target $l_\infty$ PGD attack method to adversarial examples generated by PGD (assuming true class, victim model, and attack budget are known). We tried various PGD settings (PGD10$_{\epsilon=10/255}$ refers to PGD attack using 10 steps and $\epsilon = 10/255$). Eventually, we compare these results to our CDD-RED method in Table A1.

Given that the average reconstruction error between $\mathbf{x}$ and $\mathbf{x}'$ is 20.60, we can see from Table A1 that PGD attacks further enlarge the distortion from the clean data. Although PGD attacks can achieve high accuracy after reverting the adversarial data back to their true labels, the resulting perturbation estimate is far from the ground-truth in terms of their prediction alignment. We can tell from the low PA$_{\mathrm{adv}}$ by PGD methods that $\mathbf{x}'_{\mathrm{RED}}$ does not align with the input $\mathbf{x}'$ at all.

Table A1: The performance comparison among DO, DS, and CDD-RED on the CIDAR-10 dataset.

|  | PGD10$\epsilon_{20/255}$ | PGD10$\epsilon_{10/255}$ | PGD20$\epsilon_{20/255}$ | CDD-RED |
|---|---|---|---|---|
| $d(\mathbf{x}, \mathbf{x}_{\mathrm{RED}})$ | 27.63 | 22.67 | 27.53 | **11.73** |
| PA$_{\mathrm{benign}}$ | 96.20% | 82.60% | **99.80%** | 83.20% |
| PA$_{\mathrm{adv}}$ | 6.20% | 7.20% | 4.80% | **97.40%** |

## B  DATASET DETAILS.

The training and testing dataset is composed of three attack methods including PGD Madry et al. (2017), FGSM Goodfellow et al. (2014), and CW attack Carlini & Wagner (2017), applied to **5 models** including pre-trained ResNet18 (Res18), Resnet50 (Res50) He et al. (2015), VGG16, VGG19, and InceptionV3 (IncV3) Szegedy et al. (2015). By default, PGD attack and FGSM attack are bounded by $\ell_\infty$-norm constraint and CW attack is bounded by $\ell_2$-norm. The range of the perturbation strength $\epsilon$ for PGD attack and FGSM attack are $[1/255, 40/255)$ and $[1/255, 20/255)$, respectively. As for CW attack, the adversary's confidence parameter $k$ is uniformly sampled from from $[1, 20]$. One attack method is applied to one victim model to obtain **3K successfully attacked images**. As a consequence, 45K ($3 \times 5 \times 3$K) adversarial images are generated in total: 37.5K for training and 7.5K for validating. The testing set contains 28K adversarial images generated with the same attack method & victim model.

## C  PERFORMANCE ON MORE ATTACK TYPES

We show more evaluations of the RED approaches on unforeseen attacks during training. The denoisers are all trained on the training dataset containing adversarial examples generated on the ImageNet dataset, as in Appendix B. The test data includes adversarial examples generated on the CIFAR-10 dataset in C.1, Wasserstein minimized attackers in C.2, and attacks on smoothed classifiers in C.3.

### C.1  PERFORMANCE ON CIFAR-10 DATASET

We further evaluate the performance of the RED approaches on the adversarial examples generated on the CIFAR-10 dataset. As the denoiser is input agnostic, we directly test the denoiser trained on adversarial examples generated on the ImageNet dataset. Here we consider the 10-step PGD-$l_{\mathrm{inf}}$ attack generation method with the perturbation radius $\epsilon = 8/255$. And these examples are not seen during our training. As shown in the Table A2, the proposed CDD-RED method provides the best

PA$_{\text{clean}}$ and PA$_{\text{adv}}$ with a slightly larger $d(\mathbf{x}, \mathbf{x}_{\text{RED}})$ than DO. This is not surprising as DO focuses only on the pixel-level denoising error metric. However, as illustrated in Sec. 3, the other metric like PA also plays a key role in evaluating the RED performance.

Table A2: The performance comparison among DO, DS, and CDD-RED on the CIDAR-10 dataset.

|  | DO | DS | CDD-RED |
|---|---|---|---|
| $d(\mathbf{x}, \mathbf{x}_{\text{RED}})$ | **0.94** | 4.50 | 1.52 |
| PA$_{\text{benign}}$ | 9.90% | 71.75% | **71.80%** |
| PA$_{\text{adv}}$ | 92.55% | 89.70% | **99.55%** |

## C.2 PERFORMANCE ON WASSERSTEIN MINIMIZED ATTACKERS

We further show the performance on Wasserstein minimized attackers, which is an unseen attack type during training. The adversarial examples are generated on the ImageNet sub-dataset using Wasserstein ball. We follow the same setting from Wong et al. (2019), where the attack radius $\epsilon$ is 0.1 and the maximum iteration is 400 under $l_\infty$ norm inside Wasserstein ball. The results are shown in Table A3. As we can see, Wasserstein attack is a more challenging attack type for RED than the $l_p$ attack types considered in the paper, justified by the lower prediction alignement PA$_{\text{benign}}$ across all methods. This implies a possible limitation of supervised training over ($l_2$ or $l_\infty$) attacks. One simple solution is to expand the training dataset using more diversified attacks (including Wasserstein attacks). However, we believe that the further improvement of the generalization ability of RED deserves a more careful study in the future, e.g., an extension from the supervised learning paradigm to the (self-supervised) pre-training and finetuning paradigm.

Table A3: The performance comparison among DO, DS, and CDD-RED on Wasserstein minimized attackers.

|  | DO | DS | CDD-RED |
|---|---|---|---|
| $d(\mathbf{x}, \mathbf{x}_{\text{RED}})$ | **9.79** | 17.38 | 11.66 |
| PA$_{\text{benign}}$ | 92.50% | 96.20% | **97.50%** |
| PA$_{\text{adv}}$ | 35.00% | 37.10% | **37.50%** |

## C.3 PERFORMANCE ON ATTACKS AGAINST SMOOTHED CLASSIFIERS

We further show the performance on the attack against smoothed classifiers, which is an unseen attack type during training. A smoothed classifier predicts any input $x$ using the majority vote based on randomly perturbed inputs $\mathcal{N}(x, \sigma^2 I)$ Cohen et al. (2019). Here we consider the 10-step PGD-$\ell_\infty$ attack generation method with the perturbation radius $\epsilon = 20/255$, and $\sigma = 0.25$ for smoothing. As shown in Table A4, the proposed CDD-RED method provides the best PA$_{\text{clean}}$ and PA$_{\text{adv}}$ with a slightly larger $d(\mathbf{x}, \mathbf{x}_{\text{RED}})$ than DO.

Table A4: The performance comparison among DO, DS, and CDD-RED on the PGD attack against smoothed classifiers.

|  | DO | DS | CDD-RED |
|---|---|---|---|
| $d(\mathbf{x}, \mathbf{x}_{\text{RED}})$ | **15.53** | 22.42 | 15.89 |
| PA$_{\text{benign}}$ | 68.13% | 70.88% | **76.10%** |
| PA$_{\text{adv}}$ | 58.24% | 58.79% | **61.54%** |

# D COMPUTATION COST OF RED

We measure the computation cost on a single RTX Titan GPU. The inference time for DO, DS, and CDD-RED is similar as they use the same denoiser architecture. For the training cost, the maximum

training epoch for each method is set as 300. The average GPU time (in seconds) of one epoch for DO, DS, and CDD-RED is 850, 1180, and 2098, respectively. It is not surprising that CDD-RED is conducted over a more complex RED objective. Yet, the denoiser only needs to be trained once to reverse-engineer a wide variety of adversarial perturbations, including those unseen attacks during the training.

# E ABLATION STUDIES ON CDD-RED

In this section, we present additional experiment results using the proposed CDD-RED method for reverse engineering of deception (RED). We will study the effect of the following model/parameter choice on the performance of CDD-RED: 1) pretrained classifier $\hat{f}$ for PA regularization, 2) data augmentation strategy $\check{\mathcal{T}}$ for PA regularization, and 3) regularization parameter $\lambda$ that strikes a balance between the pixel-level reconstruction error and PA in (4). Recall that the CDD-RED method in the main paper sets $\hat{f}$ as VGG19, $\check{\mathcal{T}} = \{t \in \mathcal{T} \mid \hat{F}(t(\mathbf{x})) = \hat{F}(\mathbf{x}), \hat{F}(t(\mathbf{x}')) = \hat{F}(\mathbf{x}')\}$, and $\lambda = 0.025$.

## E.1 PRETRAINED CLASSIFIER $\hat{f}$.

Table A5: The performance of CDD-RED using a different pretrained classifier $\hat{f}$ (either Res50 or R-Res50) compared with the default setting $\hat{f}$ =VGG19.

| | $\hat{f}$=Res50 | $\hat{f}$=R-Res50 |
|---|---|---|
| $d(\mathbf{x}, \mathbf{x}_{\mathrm{RED}})$ ($\downarrow$ is better) | 12.84 ($\downarrow$ 0.20) | 10.09 ($\downarrow$ 2.95) |
| $\mathrm{PA_{benign}}$ ($\uparrow$ is better) | 84.33% ($\downarrow$1.38%) | 57.88% ($\downarrow$ 27.83%) |
| $\mathrm{PA_{adv}}$ ($\uparrow$ is better) | 79.94% ($\downarrow$ 0.49%) | 71.02% ($\downarrow$ 9.40%) |

Besides setting $\hat{f}$ as VGG19, Table A5 shows the RED performance using the other pretrained models, *i.e.*, Res50 and R-Res50. As we can see, the use of Res50 yields the similar performance as VGG19. Although some minor improvements are observed in terms of pixel level reconstruction error, the PA performance suffers a larger degradation. Compared to Res50, the use of an adversarially robust model R-Res50 significantly hampers the RED performance. That is because the adversarially robust model typically lowers the prediction accuracy, it is not able to ensure the class-discriminative ability in the non-adversarial context, namely, the PA regularization performance.

## E.2 DATA SELECTION FOR PA REGULARIZATION.

Table A6: Ablation study on the selection of $\check{\mathcal{T}}$ ($\check{\mathcal{T}} = \mathcal{T}$ and without (w/o) $\check{\mathcal{T}}$)) for training CDD-RED, compared with $\check{\mathcal{T}} = \{t \in \mathcal{T} \mid \hat{F}(t(\mathbf{x})) = \hat{F}(\mathbf{x}), \hat{F}(t(\mathbf{x}')) = \hat{F}(\mathbf{x}').\}$

| | $\check{\mathcal{T}} = \mathcal{T}$ | w/o $\check{\mathcal{T}}$ |
|---|---|---|
| $d(\mathbf{x}, \mathbf{x}_{\mathrm{RED}})$ ($\downarrow$ is better) | 15.52 ($\uparrow$ 2.48) | 13.50 ($\uparrow$ 0.46) |
| $\mathrm{PA_{benign}}$ ($\uparrow$ is better) | 83.64% ($\downarrow$ 2.07%) | 84.04% ($\downarrow$1.67%) |
| $\mathrm{PA_{adv}}$ ($\uparrow$ is better) | 75.92% ($\downarrow$ 4.51%) | 79.99% ($\downarrow$ 0.44%) |

As data augmentation might alter the classifier's original decision in (4), we study how $\check{\mathcal{T}}$ affects the RED performance by setting $\check{\mathcal{T}}$ as the original data, i.e., without data augmentation, and all data, i.e., $\check{\mathcal{T}} = \mathcal{T}$. Table A6 shows the performance of different $\check{\mathcal{T}}$ configurations, compared with the default setting. The performance is measured on the testing dataset. As we can see, using all data or original

data cannot provide an overall better performance than CDD-RED. That is because the former might cause over-transformation, and the latter lacks generalization ability.

### E.3 REGULARIZATION PARAMETER $\lambda$.

Table A7: Ablation study on the regularization parameter $\lambda$ (0, 0.0125, and 0.05) for CDD-RED training, compared with $\lambda$=0.025.

| | $\lambda$=0 | $\lambda$=0.0125 | $\lambda$=0.05 |
|---|---|---|---|
| $d(\mathbf{x}, \mathbf{x}_{\text{RED}})$ ($\downarrow$ is better) | 8.92 ($\downarrow$ 4.12) | 12.79 ($\downarrow$ 0.25) | 14.85 ($\uparrow$ 2.13) |
| $\text{PA}_{\text{benign}}$ ($\uparrow$ is better) | 47.61% ($\downarrow$38.10%) | 81.00% ($\downarrow$4.71%) | 85.56% ($\downarrow$ 0.15%) |
| $\text{PA}_{\text{adv}}$ ($\uparrow$ is better) | 73.37% ($\downarrow$7.06%) | 78.25% ($\downarrow$ 2.18%) | 79.94% ($\downarrow$ 0.49%) |

The overall training objective of CDD-RED is (4), which is the weighted sum of the reconstruction error and PA with a regularization parameter $\lambda$. We further study the sensitivity of CDD-RED to the choice of $\lambda$, which is set by $0, 0.0125$, and $0.05$. Table A7 shows the RED performance of using different $\lambda$ values, compared with the default setting $\lambda = 0.025$. We report the average performance on the testing dataset. As we can see, the use of $\lambda = 0$, which corresponds to training the denoiser without PA regularization, achieves a lower pixel-level reconstruction error $d(\mathbf{x}, \mathbf{x}_{\text{RED}})$, but degrades the prediction-level performance, especially $\text{PA}_{\text{benign}}$ greatly. In the same time, $\lambda = 0$ provides a smaller pixel-level reconstruction error with a better PA performance than DO, which indicates the importance of using proper augmentations. We also observe that keep increasing $\lambda$ to $0.05$ is not able to provide a better PA.

## F  ABLATION STUDY OF DIFFERENT ATTACK HYPERPARAMETER SETTINGS

We test on PGD attacks generated with different step sizes, including $4/255$ and $6/255$, and with and without random initialization (RI). Other hyperparameters are kept the same. The adversarial examples are generated by the same set of images w.r.t. the same classifier ResNet-50. The results are shown in Table A8. As we can see, the RED performance is quite robust against the varying hyperparameters of PGD attacks. Compared with DO, CDD-RED greatly improves $\text{PA}_{\text{benign}}$ and achieves higher $\text{PA}_{\text{adv}}$ with a slightly larger $d(\mathbf{x}, \mathbf{x}_{\text{RED}})$. Compared to DS, CDD-RED achieves slightly better PA but with a much smaller reconstruction error $d(\mathbf{x}, \mathbf{x}_{\text{RED}})$.

Table A8: RED performance for PGD with different hyperparameter settings, including stepsize as $4/255$ and $6/255$, and with and without RI.

| Without RI / With RI | Stepsize | DO | DS | CDD-RED |
|---|---|---|---|---|
| $d(\mathbf{x}, \mathbf{x}_{\text{RED}})$ | 4/255 | **5.94/5.96** | 16.56/16.57 | 8.91/8.94 |
| | 6/255 | **5.99/5.98** | 16.52/16.50 | 8.97/8.94 |
| $\text{PA}_{\text{benign}}$ | 4/255 | 40.00%/47.00% | 91.00%/91.00% | **94.00%/93.00%** |
| | 6/255 | 51.00%/61.00% | 94.00%/93.00% | **95.00%/94.50%** |
| $\text{PA}_{\text{adv}}$ | 4/255 | 97.50%/97.50% | 97.50%/97.50% | **99.50%/99.50%** |
| | 6/255 | 96.50%/96.50% | 95.50%/95.50% | **98.50%/99.50%** |

## G  PERFORMANCE WITHOUT ADVERSARIAL DETECTION ASSUMPTION

The focus of RED is to demonstrate the feasibility of recovering the adversarial perturbations from an adversarial example. However, in order to show the RED performance on the global setting, we experiment with a mixture of 1) adversarial images, 2) images with Gaussian noise (images with random perturbations), and 3) clean images on the ImageNet dataset. The standard deviation of

the Gaussian noise is set as 0.05. Each type of data accounts for **1/3** of the total data. The images are shuffled to mimic the live data case. The overall accuracy before denoising is 63.08%. After denoising, the overall accuracy obtained by DO, DS, and CDD-RED is 72.45%, 88.26%, and **89.11%**, respectively. During the training of the denoisers, random noise is not added to the input.

## H   TRANSFERABILITY OF RECONSTRUCTED ADVERSARIAL ESTIMATE

We further examine the transferability of RED-synthesized perturbations. In the experiment, RED-synthesized perturbations are generated from PGD attacks using ResNet-50. We then evaluate the attack success rate (ASR) of the resulting perturbations transferred to the victim models ResNet-18, VGG16, VGG19, and Inception-V3. The results are shown in Table A9. As we can see, the perturbation estimates obtained via our CDD-RED yield better attack transferability than DO and DS. Therefore, such RED-synthesized perturbations can be regarded as an efficient transfer attack method.

Table A9: The tranferability of RED-synthesized perturbations.

|  | DO | DS | CDD-RED |
|---|---|---|---|
| ResNet-18 | 66.50% | 70.50% | **77.50%** |
| VGG16 | 71.50% | 74.00% | **81.00%** |
| VGG19 | 71.50% | 70.00% | **80.00%** |
| Inception-V3 | 86.00% | 85.50% | **90.00%** |

## I   POTENTIAL APPLICATIONS OF RED

In this paper, we focus on recovering attack perturbation details from adversarial examples. But in the same time, the proposed RED can be leveraged for various potential interesting applications. In this section, we delve into three applications, including RED for adversarial detection in I.1, inferring the attack identity in I.2, and provable defense in I.3.

### I.1   RED FOR ADVERSARIAL DETECTION

The outputs of RED can be looped back to help the design of adversarial detectors. Recall that our proposed RED method (CDD-RED) can deliver an attribution alignment test, which reflects the sensitivity of input attribution scores to the pre-RED and post-RED operations. Thus, if an input is an adversarial example, then it will cause a high attribution dissimilarity (i.e., misalignment) between the pre-RED input and the post-RED input, i.e., $I(x, f(x))$ vs. $I(D(x), f(D(x)))$ following the notations in Section 3. In this sense, attribution alignment built upon $I(x, f(x))$ and $I(D(x), f(D(x)))$ can be used as an adversarial detector. Along this direction, we conducted some preliminary results on RED-assisted adversarial detection, and compared the ROC performance of the detector using CDD-RED and that using denoised smoothing (DS). In Figure A1, we observe that the CDD-RED based detector yields a superior detection performance, justified by its large area under the curve. Here the detection evaluation dataset is consistent with the test dataset in the evaluation section of the paper.

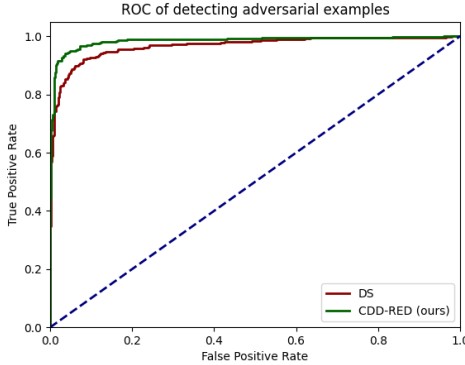

Figure A1: RoC of detecting adversarial examples.

## I.2 RED FOR ATTACK IDENTITY INFERENCE

We consider another application to infer the attack identity using the reverse-engineered adversarial perturbations. Similar to the setup of Figure 8, we achieve the above goal using the correlation screening between the new attack and the existing attack type library. Let $z'$ (e.g., PGD attack generated under the unseen AlexNet victim model) be the new attack. We can then adopt the RED model $D(\cdot)$ to estimate the perturbations $\delta_{new} = z' - D(z')$. And let $x'_{Atk_i}$ denote the generated attack over the estimated benign data $D(z')$ but using the existing attack type i. Similarly, we can obtain the RED-generated perturbations $\delta_i = x'_{Atk_i} - D(x'_{Atk_i})$. With the aid of $\delta_{new}$ and $\delta_i$ for all $i$, we infer the most similar attack type by maximizing the cosine similarity: $i^* = \mathrm{argmax}_i\, cos(\delta_{new}, \delta_i)$. Figure A2 shows an example to link the new AlexNet-generated PGD attack with the existing VGG19-generated PGD attack. The reason is elaborated on below. (1) Both attacks are drawn from the PGD attack family. And (2) in the existing victim model library (including ResNet, VGG, and InceptionV3), VGG19 has the most similar architecture as AlexNet, both of which share a pipeline composed of convolutional layers following fully connected layers without residual connections.

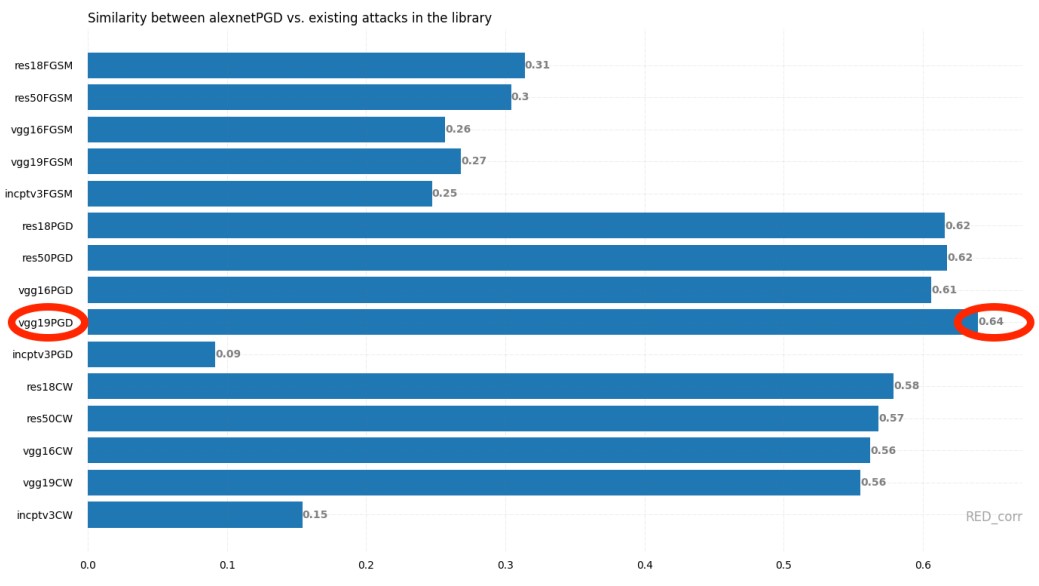

Figure A2: Similarity between AlexNet-generated PGD vs. existing attacks in the library.

### I.3 RED FOR PROVABLE DEFENSE

We train the RED models to construct smooth classifiers, the resulting certified accuracy is shown in Figure A3. Here the certified accuracy is defined by the ratio of correctly-predicted images whose certified perturbation radius is less than the $\ell_2$ perturbation radius shown in the x-axis.

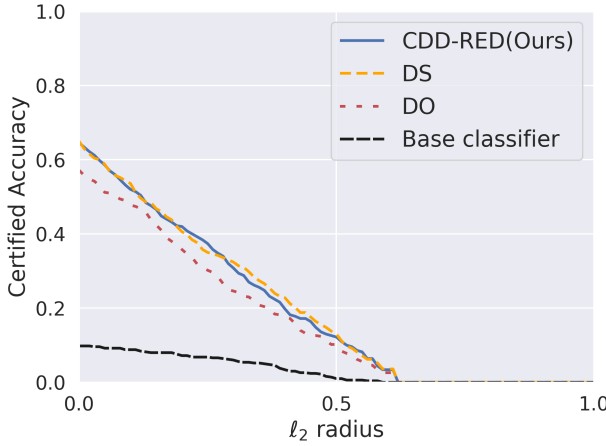

Figure A3: Certified Robustness by different methods.

