# OpenReview forum: "Reverse Engineering of Imperceptible Adversarial Image Perturbations"
_ICLR.cc/2022/Conference — ICLR 2022 Poster_

### Official Review · Reviewer_e8YB · 2021-10-31

**Correctness:** 4
**Technical Novelty And Significance:** 2
**Empirical Novelty And Significance:** 3
**Recommendation:** 6
**Confidence:** 4

**Main Review:**

Strengths:
1. The problem considered here is relatively less explored. The paper is well-organized, easy to read, and the algorithmic approach is sound. Overall the paper should be accessible to a wider audience.

2. The experimental results show that the new architecture is more effective than the baseline approaches for the RED problem considered in this paper.

Weaknesses:

1. The authors do not provide any concrete applications of RED. They mention that the estimated REDs can be used for diagnosing the model in a forensic manner. But it is not clear what one can get out of such type of analysis. One natural application is using REDs for defending against adversarial examples. Given that one of the baselines Denoised Smoothing (DS) that the authors compare against can be used for constructing a provable defense, I am not sure why the RED approach cannot be leveraged to build some sort of defense. Another application can be using REDs for explainability,i.e., to perhaps locate blind spots/missing data in the training set.

2. While the problem studied here is now, the method is somewhat straightforward and consists of combining two architectures via adding their respective losses. Further, unlike DS it appears that estimating the REDs cannot provide any theoretical guarantees.


More questions:

1. Have you seen any significant differences in the distribution of the REDs generated by the different attacks, e.g., if certain perturbations can be "only" generated by a certain attack?

2. How does the method compare with the baselines on estimating the REDs on smoothed classifiers?

3. Will the method perform as well on smaller datasets, e.g., MNIST or CIFAR10? Similarly, can REDs be generalized to non-image domains?

4. The text mentions that "Based on the correlation screening, it can infer the properties of the new attack type based on its most similar counterpart in the existing attack library", what can a user gain from inferring these properties?

5. Given that many adversarial attacks are sensitive to the hyperparameters, e.g., initialization, step size for PGD etc., how does the alignment and reconstruction error vary on the different examples created by the same method on the same image wrt the same classifier?

6. Can one build a library of adversarial perturbations collected from different sources and then to test a new model use those instead of running adversarial attacks? Will that provide some sort of reliability?


**Summary Of The Paper:**

The paper considers the problem of automatically reconstructing adversarial perturbations from examples in a post-hoc manner. The authors argue that for an effective reconstruction, it is not sufficient to only minimize the reconstruction error but also it is essential to align the predictions of the original and their reconstructed versions. To achieve these goals, the authors combine a denoising network, with a prediction alignment network via a standard combination of their respective losses. The authors incorporate data augmentation to further improve the performance of their approach. The empirical result indicates that the new architecture is able to better balance the prediction alignment and the reconstruction error than the baselines.

**Summary Of The Review:**

I believe that the paper explores an interesting problem but does not show any relevant practical applications for an end-user who might be interested in improving the robustness, interpretability, or fairness of their models before/as they are deployed in the real world. I encourage the authors to study which parts of the deployment pipeline can benefit from the estimated REDs.

---

> ### Author Response · Authors · 2021-11-21
> **Response to Reviewer e8YB (Part V)**
>
> **Q8. The text mentions that "Based on the correlation screening, it can infer the properties of the new attack type based on its most similar counterpart in the existing attack library", what can a user gain from inferring these properties? The paper does not show any relevant practical applications for an end-user who might be interested in improving the robustness, interpretability, or fairness of their models before/as they are deployed in the real world.**
>
> **A8:** Thank you very much for your comments. We have responded to this question as a part of  **A1**. We highlight our previous response below.
>
>
> **Recovery of attack type correlation** ***[(Fig.8)](https://ibb.co/KbwW4bQ)***: The insight from this example is that suppose one has an inventory of existing attack types. If a new attack appears, then one can resort to RED to estimate the correlations between the new attack type and the existing attack types. Based on the correlation screening, it can infer the details of the new attack type based on its most similar counterpart in the existing attack library. In the next, we show that this idea can be used for inference of new attack details (e.g., the victim model used for attack generation).
>
> Similar to the setup of Fig. 8, we achieve the goal of attack identity inference using the correlation screening between the new attack and the existing attack type library. Let $z^\prime$  (e.g., PGD attack generated under the unseen AlexNet victim model) be the new attack. We can then adopt  the RED model $D(\cdot)$ to estimate the perturbations $\delta_{new} = z^\prime - D(z^\prime)$. And let $x^\prime_{Atk_i}$ denote the generated attack over the estimated benign data $D(z^\prime)$ but using the existing attack type $i$. Similarly, we can obtain the RED-generated perturbations $\delta_{i} = x^\prime_{Atk_i} - D(x^\prime_{Atk_i})$. With the aid of $\delta_{new}$ and $\delta_{i}$ for all $i$, we infer the most similar attack type by maximizing the cosine similarity: $i^* = argmax_{i} \ cos(\delta_{new},\delta_{i})$. [Figure](https://ibb.co/zRtNjgY) shows an example to link the new AlexNet-generated PGD attack with the existing VGG19-generated PGD attack. The reason is elaborated on below. (1) Both attacks are drawn from the PGD attack family. And (2) in the existing victim model library (including ResNet, VGG, and InceptionV3), VGG19 has the most similar architecture as AlexNet, both of which share a pipeline composed of convolutional layers following fully connected layers without residual connections.
>
>
>
>
> **Q9. Can one build a library of adversarial perturbations collected from different sources and then test a new model using those instead of running adversarial attacks? Will that provide some sort of reliability?**
>
> **A9:** Thank you very much for the insightful comment. To the best of our understanding, the comment suggested using RED to build a library of adversarial perturbations so that the adversarial attack against a new model can be achieved using the existing perturbation instance in the library without calling an optimization-based attack generation method. Please feel free to let us know if we misunderstood the comment.
>
> By doing the suggested approach, then a transfer attack setting is present as the RED-synthesized perturbations need to transfer to the new model. Compared to the optimization-based attack method, the transfer attack should yield a lower attack success rate (ASR). In this sense,  the use of a pre-collected perturbation library may hamper the attack performance. On the other hand, the reviewer's comment inspires us to examine the transferability of RED-synthesized perturbations. To this end, we conducted a new experiment in which RED-synthesized perturbations are generated from PGD attacks using **ResNet-50**. We then evaluate the ASR of the resulting perturbations transferred to the victim models **ResNet-18**, **VGG16**, **VGG19**, and **Inception-V3**. The results are shown in the table below. As we can see, the perturbation estimates obtained via our **CDD-RED** yield **better attack transferability** than DO and DS.
>
>
>
> |    Attack Suc. Rate      |    DO    |   DS   | CDD-RED (ours)|
> | -------- | -------- |--------|--------|
> |  ResNet-18 |  66.50\%  | 70.50\%   | **77.50\%**    |
> | VGG16   |  71.50\% | 74.00\%   | **81.00\%**   |
> | VGG19   |  71.50\%  | 70.00\%   | **80.00\%**   |
> | Inception-V3   | 86.00\%  | 85.50\%   | **90.00\%**   |

---

> > ### Comment · Reviewer_e8YB · 2021-11-28
> > **Thanks for the detailed response**
> >
> > Dear authors,
> >
> > Thanks for the impressive response and adding experiments demonstrating new applications. While the novelty of the work is limited, the proposed concept is new and the authors now show new exciting applications. I am happy to increase my score to 6, I encourage the authors to provide detailed experimental settings for the new examples added in the paper.

---

> > > ### Author Response · Authors · 2021-11-28
> > > **Thank you!**
> > >
> > > Dear Reviewer e8YB,
> > >
> > > Thank you very much for the follow-up comment and for raising the score. We will be sure to follow your suggestion to include detailed application settings, results and discussion in the updated version.
> > >
> > > Thanks,

---

> ### Author Response · Authors · 2021-11-21
> **Response to Reviewer e8YB (Part IV)**
>
> **Q6: The generalization to smaller dataset.**
>
> **A6:** This is a very good suggestion. Thus, we conducted **new experiments** to evaluate the performance of the RED approaches on the adversarial examples generated on the **CIFAR-10** dataset. Here we consider the $10$-step PGD-$l_\inf$ attack generation method with the perturbation radius $\epsilon = 8/255$. And these examples are not seen during our training. As shown in the following table, the proposed **CDD-RED** method provides the **best** $PA_{\text{clean}}$ and $PA_{\text{adv}}$ with a slightly larger $d(\mathbf x, \mathbf x_\mathrm{RED})$ than DO. This is not surprising as DO focuses only on the pixel-level denoising error metric. However, as illustrated in Sec. 3, the other metric like prediction alignment (PA) also plays a key role in evaluating the RED performance.
>
> |    RED on CIFAR-10      |    DO    |   DS   | CDD-RED (ours)|
> | -------- | -------- |--------|--------|
> |  $\text{PA}_{\text{adv}}$ | 92.55\%  | 89.70\%   | **99.55\%** |
> |  $\text{PA}_{\text{benign}}$ | 9.90\% | 71.75\% | **71.80\%**    |
> | $d(\mathbf x, \mathbf x_\mathrm{RED})$ | **0.94** | 4.50| 1.52   |
>
>
> **Q7: How does the alignment and reconstruction error vary on the examples with different hyperparameters including initialization and step size for PGD?**
>
> **A7:** We test on PGD attacks generated with different step sizes, including $4/255$ and $6/255$, and with and without random initialization (RI). Other hyperparameters are kept the same. The adversarial examples are generated by the same set of images w.r.t. the same classifier ResNet-50. As we can see, the RED performance is quite robust against the varying hyperparameters of PGD attacks. Compared with DO, **CDD-RED** greatly improves $PA_{\text{benign}}$ and achieves higher $PA_{\text{adv}}$ with a slightly larger $d(\mathbf x, \mathbf x_\mathrm{RED})$. Compared to DS, **CDD-RED** achieves slightly better prediction alignment (PA) but with a much smaller reconstruction error $d(\mathbf x, \mathbf x_\mathrm{RED})$.
>
>
>
>
>
>
> |    Stepsize = 4/255: without RI / with RI|    DO    |   DS   | CDD-RED (ours)|
> | -------- | -------- |--------|--------|
> |  $\text{PA}_{\text{adv}}$ | 97.50\%/97.50\%     | 97.50\%/97.50\%   | **99.50\%/99.50\%**    |
> |  $\text{PA}_{\text{benign}}$ | 40.00\%/47.00\%    | 91.00\%/91.00\%  | **94.00\%/93.00\%**    |
> | $d(\mathbf x, \mathbf x_\mathrm{RED})$   | **5.94 /5.96**   | 16.56/16.57   | 8.91/8.94   |
>
> |    Stepsize = 6/255: without RI / with RI     |    DO    |   DS   | CDD-RED (ours)|
> | -------- | -------- |--------|--------|
> |  $\text{PA}_{\text{adv}}$ | 96.50\%/96.50\%    | 95.50\%/95.50\%   | **98.50\%/99.50\%**   |
> |  $\text{PA}_{\text{benign}}$ | 51.00\%/61.00\%     |94.00\%/93.00\%   | **95.00\%/94.50\%**    |
> | $d(\mathbf x, \mathbf x_\mathrm{RED})$   | **5.99/5.98**    | 16.52/16.50   | 8.97/8.94   |

---

> ### Author Response · Authors · 2021-11-21
> **Response to Reviewer e8YB (Part III)**
>
> **Q3: Given that one of the baselines Denoised Smoothing (DS) that the authors compare against can be used for constructing a provable defense, I am not sure why the RED approach cannot be leveraged to build some sort of defense. Unlike DS it appears that estimating the REDs cannot provide any theoretical guarantees.**
>
> **A3:** Thank you for the detailed comments on the application of RED to provable defense. Please see our response below.
>
> First, as we replied in **Q1**, our RED approach is not shaped to provide a provable defense. However, with simple modifications, it can also be used as a provable defense. Note that both DS and our RED approach (CDD-RED) can yield robustness guarantees only if **smooth classification** is applied to their prepended classifiers.
>
> Second, following the reviewer's suggestion, we conducted a new experiment to train the RED models to construct smooth classifiers, the resulting certified accuracies are shown in [Figure](https://ibb.co/cy5yYCL). Here the certified accuracy is defined by the ratio of correctly-predicted images whose certified perturbation radius is larger than the $\ell_2$ perturbation radius shown in the x-axis.
>
>
> **Q4: Have you seen any significant differences in the distribution of the REDs generated by the different attacks, e.g., if certain perturbations can be "only" generated by a certain attack?**
>
> **A4:** Yes, [Fig.8](https://ibb.co/KbwW4bQ) in our submission revealed this phenomenon. The correlation map constructed between the ground-truth adversarial perturbations and the reverse-engineered perturbations provides the following observations: If different attacks (built upon different attack methods and victim models) are considered, then RED can characterize such difference, justified by the block diagonal structure of the correlation map.
>
>
> **Q5: How does the method compare with the baselines on estimating the REDs on smoothed classifiers?**
>
> **A5:** This is a very good suggestion. Thus, we conducted **new experiments** to evaluate the performance of the RED approaches on the adversarial examples generated against the **smoothed classifiers**. A smoothed classifier makes a majority-vote-type prediction using the randomly perturbed inputs $\mathcal{N}(x,\sigma^2I)$ (around the original input $x$)  [1]. Here we consider the 10-step PGD-$\ell_\infty$ attack generation method with the perturbation radius $\epsilon=20/255$, and $\sigma=0.25$ for smoothing. As shown in the following table, the proposed **CDD-RED** method provides the **best** $\text{PA}_\text{clean}$ and $\text{PA}_\text{adv}$ with a slightly larger $d(\mathbf x, \mathbf x_\mathrm{RED})$ than DO.
>
>
> |    RED on smoothed classifier      |    DO    |   DS   | CDD-RED (ours)|
> | -------- | -------- |--------|--------|
> |  $\text{PA}_{\text{adv}}$ | 58.24\%  | 58.79\%   | **61.54\%** |
> |  $\text{PA}_{\text{benign}}$ | 68.13\% | 70.88\% | **76.10\%**   |
> | $d(\mathbf x, \mathbf x_\mathrm{RED})$ | **15.53** | 22.42| 15.89   |
>
>
> [1] Cohen, J. M., Rosenfeld, E., and Kolter, J. Z. Certified adversarial robustness via randomized smoothing. arXiv preprint arXiv:1902.02918, 2019.

---

> ### Author Response · Authors · 2021-11-21
> **Response to Reviewer e8YB (Part II)**
>
>
> **2. [New experiments: Applications of RED inspired from (1-2A) and (1-2B)]**
>
>
> **First**, spurred by (1-2A), we leverage the outputs of RED to help the design of adversarial detectors. Recall that our proposed RED method (CDD-RED) can deliver an **attribution alignment** test, which reflects the sensitivity of input attribution scores to the pre-RED and post-RED operations. Thus, if an input is an adversarial example, then it will cause a high attribution dissimilarity (i.e., misalignment) between the pre-RED input and the post-RED input, i.e., $I(x, f(x))$ vs. $I(D(x), f(D(x)))$ following the notations in Sec.3. In this sense, attribution alignment built upon $I(x, f(x))$ and $I(D(x), f(D(x)))$ can be used an adversarial detector. Along this direction, we conducted some preliminary results on RED-assisted adversarial detection, and compared the ROC performance of the detector using CDD-RED and that using denoised smoothing (DS).  In [figure](https://ibb.co/JQ1NWJ2), we observe that the CDD-RED based detector yields a superior detection performance, justified by its large area under the curve. Here the detection evaluation dataset is consistent with the test dataset in the evaluation section of the paper.
>
>
> **Second**, spurred by (1-2B), we consider another application to infer the attack identity using the reverse-engineered adversarial perturbations. Similar to the setup of Fig. 8, we achieve the above goal using the correlation screening between the new attack and the existing attack type library. Let $z^\prime$  (e.g., PGD attack generated under the unseen AlexNet victim model) be the new attack. We can then adopt  the RED model $D(\cdot)$ to estimate the perturbations $\delta_{new} = z^\prime - D(z^\prime)$. And let $x^\prime_{Atk_i}$ denote the generated attack over the estimated benign data $D(z^\prime)$ but using the existing attack type i. Similarly, we can obtain the RED-generated perturbations $\delta_{i} = x^\prime_{Atk_i} - D(x^\prime_{Atk_i})$. With the aid of $\delta_{new}$ and $\delta_{i}$ for all $i$, we infer the most similar attack type by maximizing the cosine similarity: $i^* = argmax_{i} \ cos(\delta_{new},\delta_{i})$. [Figure](https://ibb.co/zRtNjgY) shows an example to link the new AlexNet-generated PGD attack with the existing VGG19-generated PGD attack. The reason is elaborated on below. (1) Both attacks are drawn from the PGD attack family. And (2) in the existing victim model library (including ResNet, VGG, and InceptionV3), VGG19 has the most similar architecture as AlexNet, both of which share a pipeline composed of convolutional layers following fully connected layers without residual connections.
>
>
>
> **Q2: While the problem studied here is now, the method is somewhat straightforward and consists of combining two architectures via adding their respective losses.**
>
> **A2:** Thanks for the comment. However, we do not think that our methodology is straightforward.
>
> First, the use of both benign prediction and adversarial prediction alignment is not trivial. This, together with our prediction alignment evaluation metrics, is a key to ensuring the RED ability. In particular, the adversarial prediction alignment is defined in RED for the first time.
>
> Second, the exploration of proper data augmentation methods is also not trivial. We find that (Figure 4 in the paper) CutMix and cropping & padding can increase the pixel-level and prediction-level performance simultaneously, thus are considered as the appropriate augmentation methods to boost the RED performance. Note that the most common augmentation types like rotation and translation improve the prediction alignment of RED, but it hampers the denoising ability. Thus, data augmentation should strike a proper balance between class-discriminative ability and pixel-level denoising ability.
>
> Third, as we responded to **(Q1)**, our contributions also exist in RED formulation, evaluation metric design, and experimentation.

---

> ### Author Response · Authors · 2021-11-21
> **Response to Reviewer e8YB (Part I)**
>
> Thank you very much for the constructive comments. We address them below.
>
> **Q1: The authors do not provide any concrete applications of RED. Given that one of the baselines Denoised Smoothing (DS) that the authors compare against can be used for constructing a provable defense, I am not sure why the RED approach cannot be leveraged to build some sort of defense. Another application can be using REDs for explainability,i.e., to perhaps locate blind spots/missing data in the training set.**
>
> **A1:** Thank you very much for the detailed comments. We would like to address your concerns from the following aspects.
>
> **1. [Our focus: Formalized RED problem and solution pipeline]**
>
> **(1-1)** We would like to highlight that the study of RED is important as different from adversarial defenses, it provides a way to recover the perturbation details and understand the adversarial properties of an attack. This has been recognized by Reviewer [Ah1k](https://openreview.net/forum?id=gpp7cf0xdfN&noteId=yrd5P4oQB9): "*the methodology is demonstrated to be able to handle benign inputs well the methodology could lead to effective defenses against adversarial examples*." Furthermore, the importance of RED was also highlighted in the [DARPA RED project](https://sam.gov/opp/258cc833c18749de87aba9c129ee2205/view).
>
> **(1-2)** Given the well-motivated but unexplored RED problem, the first question that should be addressed is how to formalize the RED research pipeline, including RED formulation, methodology, evaluation metrics, and feasibility test. And this is the focus of our current work. To this end, we have made a significant effort to evaluate the performance of RED from different aspects and try to cover some application-driven examples. In particular, we would like to highlight two of them which can enable us to design more concrete applications of RED:
>
>
>    **(1-2A)** **Recovery of adversary's saliency region** ***[(Fig. 5)](https://ibb.co/s91GqJY)***. This yields the class-discriminative image regions that the adversary focused on. As will be shown later, it can also inspire us to design an **attribution-guided adversarial detection** method.
>
> **(1-2B)** **Recovery of attack type correlation** ***[(Fig. 8)](https://ibb.co/KbwW4bQ)***: The insight from this example is that suppose one has an inventory of existing attack types. If a new attack appears, then one can resort to RED to estimate the correlations between the new attack type and the existing attack types. Based on the correlation screening, it can infer the details of the new attack type based on its most similar counterpart in the existing attack library. As will be shown later, this idea can be used for inference of new attack details (e.g., the victim model used for attack generation).

---

> ### Author Response · Authors · 2021-11-24
> **Look forward to your post-rebuttal feedback!**
>
> Dear Reviewer e8YB,
>
> Thank you very much for taking the time to review our paper. We cherish your comments very much. In our earlier posted response, we have conducted new experiments and added additional clarifications to alleviate your concerns about our original submission. Please refer to our response listed at [Part I](https://openreview.net/forum?id=gpp7cf0xdfN&noteId=UlsU6qmd9qK), [Part II](https://openreview.net/forum?id=gpp7cf0xdfN&noteId=x7N29z2kMGW), [Part III](https://openreview.net/forum?id=gpp7cf0xdfN&noteId=5QiWrtN3dC2), [Part IV](https://openreview.net/forum?id=gpp7cf0xdfN&noteId=mZQJqwnXlUu), and [Part V](https://openreview.net/forum?id=gpp7cf0xdfN&noteId=tQrw8KnuCka) respectively. Please also feel free to find a summary of our response to all reviewers at [Summary Link](https://openreview.net/forum?id=gpp7cf0xdfN&noteId=XS_FHB96yJr) and highlighted general response at [General Response Link](https://openreview.net/forum?id=gpp7cf0xdfN&noteId=X4Ly-ab1yIO).
>
> We hope that you can find our effortful response convincing. If you have additional comments, please feel free to let us know. We will try our best to address them.

---

### Official Review · Reviewer_Ah1k · 2021-11-02

**Correctness:** 3
**Technical Novelty And Significance:** 4
**Empirical Novelty And Significance:** 4
**Recommendation:** 8
**Confidence:** 4

**Main Review:**

The proposed reverse engineering topic is interesting and novel. The objective seems to be a strong step beyond denoising and could be a strong tool for better understanding and defending against adversarial examples. The evaluation metrics provide a good starting point for understanding the objectives of reverse engineering adversarial perturbations. The overall methodology is straight-forward and would be practically available to both researchers and the industry.

However, the experimental results are limited to the ImageNet dataset, it is difficult to be confident that the experimental results would generalize beyond that dataset without incorporating more datasets. Adversarial examples are inputs specifically designed to fool deep learning models, if an attacker is aware of the use of this methodology would they be able to craft adversarial example to fool both the original and this model?


**Summary Of The Paper:**

This paper proposes a methodology for reverse engineering adversarial perturbations. This allows a defender to recover the original image used to produce an adversarial example and may be an effective tool to mitigating adversarial example attacks.  The paper introduces the concept of reverse engineering adversarial perturbations, defines metrics that quantify reverse engineering performance, creates a framework for training a denoising model to find the adversarial perturbations, and compares the resulting model against adversarial denoising techniques.

**Summary Of The Review:**

The major strengths of this paper are:

•	The paper is novel and could be have a considerable impact.

•	The paper provides a strong background that establishes a foundation for reverse engineering adversarial perturbations.

•	The ability to reverse engineer adversarial perturbations could be a powerful tool for understanding adversarial examples better.

•	Because the methodology is demonstrated to be able to handle benign inputs well the methodology could lead to effective defenses against adversarial examples.

The major weaknesses of the paper are:

•	The work only evaluates the ImageNet dataset. And so, it is not clear how it would generalize to other scenarios.

•	While the work compares against attacks it wasn’t trained to recognize, it appears that these methodologies represent similar attacks. (L2 or Linf) attackers It is not clear from these results that the methodology would extend to other types of attackers including L0 or Wasserstein minimized attackers.

Additional Comments:

•	The Adaptive Attack does not appear to be cited in the text.

•	The Interpretation function in the second sentence of the “Attribution alignment” subsection appears to be formatted differently than the rest of the text.

---

> ### Author Response · Authors · 2021-11-21
> **Response to Reviewer Ah1k**
>
> We sincerely appreciate your careful review and a great summary of our contributions. And thank you very much for the very constructive comments, which are addressed below.
>
> **Q1: Evaluation on other datasets**
>
> **A1:** This is a very good suggestion. Thus, we conducted **new experiments** to evaluate the performance of the RED approaches on the adversarial examples generated on the **CIFAR-10** dataset. Here we consider the $10$-step PGD-$l_\inf$ attack generation method with the perturbation radius $\epsilon = 8/255$. And these examples are not seen during our training. As shown in the following table, the proposed **CDD-RED** method provides the **best** $PA_{\text{clean}}$ and $PA_{\text{adv}}$ with a slightly larger $d(\mathbf x, \mathbf x_\mathrm{RED})$ than DO. This is not surprising as DO focuses only on the pixel-level denoising error metric. However, as illustrated in Sec. 3, the other metric like prediction alignment (PA) also plays a key role in evaluating the RED performance.
>
> |    RED on CIFAR-10      |    DO    |   DS   | CDD-RED (ours)|
> | -------- | -------- |--------|--------|
> |  $\text{PA}_{\text{adv}}$ | 92.55\%  | 89.70\%   | **99.55\%** |
> |  $\text{PA}_{\text{benign}}$ | 9.90\% | 71.75\% | **71.80\%**    |
> | $d(\mathbf x, \mathbf x_\mathrm{RED})$ | **0.94** | 4.50| 1.52   |
>
> **Q2: What if an attacker is aware of the use of this methodology**
>
> **A2:** To alleviate this concern, we evaluated the RED performance over adaptive attack (that is aware of the RED model) in **[Table 2](https://ibb.co/HhB2QTc)**. We generate the adaptive attack as below by evading the RED model but can still fool the victim model:
>
> $\min_\delta\|  \mathcal D_{\theta} (\mathbf x + \delta) - \mathbf x  \|_1  - \lambda \mathrm{CE} ( f(\mathbf x + \delta),  f (\mathbf x))$
>
> where $\mathcal D_{\theta}$ denotes the pre-trained RED model, $f$ denotes the victim model, CE denotes the cross-entropy loss, and $\lambda > 0$ is a regularization parameter to promote the prediction error when facing the perturbation $\delta$.
>
> **Q3: Extension to other types of attackers such as Wasserstein minimized attackers**
>
> **A3:** Thank you very much for the good suggestion. We added new experiments to evaluate the performance of RED models on **Wasserstein attack**, which is an unseen attack type during training. The adversarial examples are generated on the ImageNet sub-dataset using Wasserstein ball. We follow the same setting from [1], where the attack radius $\epsilon$ is 0.1 and the maximum iteration is 400 under $l_\inf$ norm inside the Wasserstein ball. The results are shown in the table below. As we can see, the Wasserstein attack is a more challenging attack type for RED than the Lp attack types considered in the paper, justified by the lower prediction alignment $\text{PA}_{\text{benign}}$ across all methods. This implies a possible limitation of supervised training over (L2 or Linf) attacks. One simple solution is to expand the training dataset using more diversified attacks (including Wasserstein attacks). However, we believe that the further improvement of the generalization ability of RED deserves a more careful study in the future, e.g., an extension from the supervised learning paradigm to the (self-supervised) pre-training + finetuning paradigm. Thank you very much for pointing this out. As a side note, even in the presence of the Wasserstein attack, our proposal yields the consistent performance shown in the paper: CDD-RED outperforms the baselines in prediction alignment (with a small margin) and has a much smaller reconstruction error than DS.
>
>
> |   RED vs. Wasserstein attack       |    DO    |   DS   | CDD-RED (ours)|
> | -------- | -------- |--------|--------|
> |  $\text{PA}_{\text{adv}}$ | 35.00\%     | 37.10\%   | **37.50\%**    |
> |  $\text{PA}_{\text{benign}}$ | 92.50\%     | 96.20\%   | **97.50\%**    |
> | $d(\mathbf x, \mathbf x_\mathrm{RED})$   |**9.79**    | 17.38   | 11.66   |
>
>
> [1]: Wong, E., Schmidt, F., and Kolter, Z. Wasserstein Adversarial Examples via Projected Sinkhorn Iterations. *In Proceedings of the 36th International Conference on Machine Learning*, pp. 6808–6817, 2019.
>
> **Q4: Additional Comments on citation of adaptive attack and inconsistent expression of “Attribution alignment”.**
>
> **A4:** Thanks for pointing out. These are fixed in the revision.

---

> > ### Comment · Reviewer_Ah1k · 2021-11-29
> > **Thank you, for the response.**
> >
> > I've read through both your response and the other reviewers comments. While I understand the concerns initially raised by the other reviewers, it is my opinion that the RED problem setting is a challenging and undeveloped research direction. And that, this work does a solid job at setting up the problem to benefit future research and proposes an initial solution to that problem. As such, I believe my original score is justified. At this point I will maintain my current score.
> >
> > It is good that the authors were able to incorporate those additional experiments. They should help in demonstrating the efficacy of the work for the readers. I do believe that readers would benefit from an even broader base of such experimental verification but that the work is sufficient in its current form.

---

> > > ### Author Response · Authors · 2021-11-29
> > > **Thank you!**
> > >
> > > Dear Reviewer Ah1k,
> > >
> > > Thanks again for your careful review and valuable comments. It is our great pleasure to see that your original rating is justified. In the updated version, we will be sure to take into account your comment 'readers would benefit from an even broader base of such experimental verification' by including some additional discussion on our newly proposed RED application setups and results.
> > >
> > > Thank you very much,
> > >
> > > Authors,

---

> ### Author Response · Authors · 2021-11-25
> **Thanks against for the constructive comments and our response**
>
> Dear Reviewer Ah1k,
>
> Thank you very much for the careful review of our submission. It was very encouraging to see your detailed and precise assessment of our contributions. We also cherish your intriguing comments. Our posted response included the additional experiments requested for evaluation on other datasets and Wasserstein attacks. We hope that our clarification and new results have largely alleviated your previous concerns. If you have additional questions, please feel free to let us know.
>
> Thanks,

---

### Official Review · Reviewer_T72q · 2021-11-03

**Correctness:** 3
**Technical Novelty And Significance:** 2
**Empirical Novelty And Significance:** 2
**Recommendation:** 6
**Confidence:** 2

**Main Review:**

Strengths:

* Paper is easy to understand
* Problem and threat model are well-formulated
* Comprehensive evaluation metrics
* Evaluations on adaptive attacks

Weaknesses:

* From figure 7d, it looks like DO is actually closer to groundtruth than RED is for p < 0.5 -- I may have missed it, but this is not really acknowledged or discussed in the paper
* It's unclear how much more computationally expensive the RED model is, compared to the DO or DS approaches
* How does one practically use this CDD-RED method? Run it on every image in the test-time / live data? In practice, Lp norm adversarial examples comprise of a small fraction of the test-time data/live, and we don't know which examples are the Lp norm adversarial examples. In this study, there are no "control" evaluations -- e.g., on images with random or no perturbations -- so it's unclear that CDD-RED will work better than baselines in the global setting.

**Summary Of The Paper:**

This paper defines a new problem, Reverse Engineering of Deceptions (RED), that aims to reconstruct the adversarial perturbation applied to a clean image based on the adversarial image. Their contributions are the following:

* RED problem formulation
* Series of "principles" to estimate the perturbations (such as class-discriminative ability and data augmentations)
* Empirical study that demonstrates the effectiveness of the authors' solution across various evaluation metrics and adversarial attacks

**Summary Of The Review:**

This paper clearly defines the RED problem and conducts experiments to show that a classifier optimized for extracting adversarial perturbations can extract such perturbations better than 2 other baselines for a variety of adversarial attacks. A main concern is that the problem setting does not seem well-motivated: in what case will we _know_ whether an image is an Lp norm adversarial image and need to extract the adversarial perturbation? What could these perturbations be used for? Does the authors' proposed method work better than performing some PGD to go back to that true class? Finally, in table 2, the d(x, x_{RED}) is actually lowest for the DO baseline, and the DO baseline has comparable PA to CDD-RED. The results for CDD-RED do not seem significantly better.

---

> ### Author Response · Authors · 2021-11-21
> **Response to Reviewer T72q (Part IV)**
>
> **Q7: DO has comparable PA to CDD-RED with the lowest $d(\mathbf x, \mathbf x_\mathrm{RED})$ in Table 2.**
>
> **A7:** DO indeed has the lowest $d(\mathbf x, \mathbf x_\mathrm{RED})$ in **Table 2**. The reason is that DO purely focuses on the pixel alignment between $\mathbf x$ and $\mathbf x_\mathrm{RED}$. However, as explained in Sec. 3, RED should be evaluated in a much more comprehensive way. For example, in Table 2, CDD-RED achieves the highest $PA_\mathtt{benign}$ and $PA_{\mathtt{adv}}$ for all of the three types of unseen attacks among the three approaches.  The performance improvement is not minor. Specifically, compared to DO, CDD-RED improves the $PA_{\mathrm{benign}}$ by **9.89\%** and **10.35\%** for AutoAttack and Feature Attack, and improves $PA_{\mathrm{adv}}$ by **36.51\%** for Feature Attack. Furthermore, as we highlighted, **Table 2** is just one part of the entire RED evaluation pipeline. By combining the results on the testing dataset shown in **Table 1** and the attribution alignments in **Figure 5** and **Figure 6**, we can observe that **DO lacks the RED ability** at the prediction and the attribution levels.

---

> ### Author Response · Authors · 2021-11-21
> **Response to Reviewer T72q (Part III)**
>
> **Q5: The main concern is that the problem setting does not seem well-motivated.**
>
> **A5:** The study of RED is well-motivated. Unlike adversarial defenses, it provides a way to recover the perturbation details and understand the adversarial properties of an attack (see illustration from Fig. 1 on page 3). This has also been recognized by Reviewer [Ah1k](https://openreview.net/forum?id=gpp7cf0xdfN&noteId=yrd5P4oQB9): “The ability to reverse engineer adversarial perturbations could be a powerful tool for understanding adversarial examples better.” Furthermore, the importance of RED was also highlighted in the [DARPA RED project](https://sam.gov/opp/258cc833c18749de87aba9c129ee2205/view).
>
> Please see our [general response](https://openreview.net/forum?id=gpp7cf0xdfN&noteId=X4Ly-ab1yIO) for more practical applications that we newly added.
>
> **Q6: In what case will we know whether an image is an Lp norm adversarial image and need to extract the adversarial perturbation? What could these perturbations be used for?**
>
> **A6:** Thanks for the comment. We would like to elaborate on our response from the following aspects.
>
> 1. [Why considering Lp attack?] We focus on Lp norm-constrained adversarial image since this is unnoticeable to human eyes, and this is a more stealthy threat model. Similar to the purpose of adversarial detection against Lp attacks, we would like to make a further step to ask if the tiny perturbation can be recovered.
>
> 2. [How to know whether is Lp an adversarial image?] As we illustrated in the paragraph next to Fig. 1, we can detect the adversarial image using existing adversarial detection methods. Even if the detector may not be perfect, our study has shown that the proposed RED approach also performs well even if the input examples are benign ones or just partially perturbed images (which may not be successful attacks; see Fig. 7), and the detection-evasion feature attacks (Table 2). Moreover, in our response **A3**, we empirically showed that even without the adversarial attack prior knowledge, we can still run the RED system to process the mixed input stream of clean data, adversarial data, and data with random noise, as suggested by the reviewer.
>
> 3. [Why are extracted perturbations useful?]
> This is an insightful comment and inspires us to deliver more practical use cases beyond the feasibility study of RED. We find that RED may drive two interesting use cases.
>
>     (3a) The outputs of RED can be looped back to help the design of adversarial detectors. Re call that our proposed RED method (CDD-RED) can deliver an **attribution alignment** test, which reflects the sensitivity of input attribution scores to the pre-RED and post-RED operations. Thus, if an input is an adversarial example, then it will cause a high attribution dissimilarity (i.e., misalignment) between the pre-RED input and the post-RED input, i.e., $I(x, f(x))$ vs. $I(D(x), f(D(x)))$ following the notations in Sec.3. In this sense, attribution alignment built upon $I(x, f(x))$ and $I(D(x), f(D(x)))$ can be used an adversarial detector. Along this direction, we conducted some preliminary results on RED-assisted adversarial detection, and compared the ROC performance of the detector using CDD-RED and that using denoised smoothing (DS).  In [figure](https://ibb.co/JQ1NWJ2), we observe that the CDD-RED based detector yields a superior detection performance, justified by its large area under the curve. Here the detection evaluation dataset is consistent with the test dataset in the evaluation section of the paper.
>
>     (3b) We consider another application to infer the attack identity using the reverse-engineered adversarial perturbations. Similar to the setup of Fig. 8, we achieve the above goal using the correlation screening between the new attack and the existing attack type library. Let $z^\prime$  (e.g., PGD attack generated under the unseen AlexNet victim model) be the new attack. We can then adopt  the RED model $D(\cdot)$ to estimate the perturbations $\delta_{new} = z^\prime - D(z^\prime)$. And let $x^\prime_{Atk_i}$ denote the generated attack over the estimated benign data $D(z^\prime)$ but using the existing attack type $i$. Similarly, we can obtain the RED-generated perturbations $\delta_{i} = x^\prime_{Atk_i} - D(x^\prime_{Atk_i})$. With the aid of $\delta_{new}$ and $\delta_{i}$ for all $i$, we infer the most similar attack type by maximizing the cosine similarity: ${i}^* = argmax_{i} \ \mathtt{cos}(\delta_{new},\delta_{i})$. [Figure](https://ibb.co/zRtNjgY) shows an example to link the new AlexNet-generated PGD attack with the existing VGG19-generated PGD attack. The reason is: (1) Both attacks are drawn from the PGD attack family. (2) in the existing victim model library (including ResNet, VGG, and InceptionV3), VGG19 has the most similar architecture as AlexNet, sharing a pipeline composed of convolutional layers following fully connected layers without residual connection.

---

> ### Author Response · Authors · 2021-11-21
> **Response to Reviewer T72q (Part II)**
>
> **Q3: How does one practically use this CDD-RED method? Run it on every image in the test-time / live data? In practice, Lp norm adversarial examples comprise a small fraction of the test-time data/live, and we don't know which examples are the Lp norm adversarial examples. In this study, there are no "control" evaluations -- e.g., on images with random or no perturbations -- so it's unclear that CDD-RED will work better than baselines in the global setting.**
>
> **A3:**  Thanks for the comment.
>
> First, our experiment (**Fig. 7**) covers the case of images with **no perturbations**. This corresponds to the case with the partially perturbed ratio $p = 0$. As we can see, our method is able to recover the original image close to the ground truth. Fig. 7 also shows that compared to reverse engineering of perturbed inputs, the case of no perturbation is not difficult for all methods. We kindly remark that this has also been recognized by Reviewer [Ah1k](https://openreview.net/forum?id=gpp7cf0xdfN&noteId=yrd5P4oQB9): "*the methodology is demonstrated to be able to handle benign inputs well the methodology could lead to effective defenses against adversarial examples*."
>
> Second, we would like to kindly remark that the **focus of RED** is to demonstrate the feasibility of recovering the adversarial perturbations from an adversarial example. Thus, we did not focus on random perturbations. However, we understand reviewer's concern in the global setting. Following the reviewer's suggestion, we show the RED performance **without adversarial detection assumption**. Specifically, we experiment with a **mixture** of **1) adversarial images**, **2) images with Gaussian noise (images with random perturbations)**, and **3) clean images** on the **ImageNet** dataset. The standard deviation of the Gaussian noise is set as 0.05. Each type of data accounts for 1/3 of the total data. The images are shuffled to mimic the **live data case**. The overall accuracy before denoising is **63.08\%**. After denoising, the overall accuracy obtained by DO, DS, and **CDD-RED** is 72.45\%, 88.26\%, and **89.11\%**, respectively. During the training of the denoisers, random noise is not added to the input. To boost the accuracy of the denoisers against randomly perturbed data, one can also introduce the term $\|  \mathcal D_{\theta} (\mathbf x + \delta) - D_{\theta}(\mathbf x )\|_1$ into the training objective in the future where $\delta\sim N(0,\sigma^2)$.
> **Q4: Comparison with performing PGD back to the true class**
>
> **A4:** This is an interesting point, however, this requires additional assumptions and may **not** be a practical RED (reverse engineering of deception) approach.
>
>
> First, since PGD is a test-time deterministic optimization approach for perturbation generation, its targeted implementation requires the **true class** of the adversarial example, which could be unknown at testing time. What is more, one has to pre-define the perturbation budget $\epsilon$ for PGD. This value is also unknown.
> Second, performing PGD back to the true class might not exactly recover the ground-truth adversarial perturbations. By contrast, its RED counterpart could be over-perturbed. To make it more convincing, we applied the target $l_\infty$ PGD attack method to adversarial examples generated by PGD (assuming true class, victim model, and attack budget are known). We tried various PGD settings ($\text{PGD10}_{\epsilon=10/255}$ refers to PGD attack using 10 steps and $\epsilon=10/255$). Eventually, we compare these results to our CDD-RED method.
>
>
> |  | $\text{PGD10}_{\epsilon=20/255}$ | $\text{PGD10}_{\epsilon=10/255}$ | $\text{PGD20}_{\epsilon=20/255}$ |  CDD-RED (ours)   |
> | -------------------------------------- | -------------------------------- | -------------------------------- | -------------------------------- | --- |
> | $\text{PA}_{\text{adv}}$               | 6.2%                             | 7.2%                             | 4.8%                             |  **97.40%**   |
> | $\text{PA}_{\text{benign}}$            | 96.2%                            | 82.6%                            | **99.8%**                            |   83.20%  |
> | $d(\mathbf x, \mathbf x_\mathrm{RED})$ | 27.63                            | 22.67                            | 27.53                            |  **11.73**   |
>
>
> Given that the average reconstruction error between $\mathbf x$ and $\mathbf x'$ is 20.60, we can see from the above table that PGD attacks further enlarge the distortion from the clean data. Although PGD attacks can achieve high accuracy after reverting the adversarial data back to their true labels, the resulting perturbation estimate is far from the ground truth in terms of their prediction alignment. We can tell from the low $PA_{\text{adv}}$  by PGD methods that $\mathbf x_{\mathrm {RED}}'$ does not align with the input $\mathbf x'$ at all.

---

> ### Author Response · Authors · 2021-11-21
> **Response to Reviewer T72q (Part I)**
>
> Many thanks to the reviewer for your constructive feedback.
>
> **Q1: From [Fig 7](https://ibb.co/pLd7bwQ)(d), it looks like DO is actually closer to the ground truth than RED is for p < 0.5 -- I may have missed it, but this is not really acknowledged or discussed in the paper**.
>
> **A1:** We apologize for the confusion. In Fig.7(d), if viewing from the reconstruction error metric, then yes, DO is closer to the ground-truth than CDD-RED (ours) as the input is partially perturbed (p < 0.4). However, we would like to make two remarks.
>
> First, DO may yield a better reconstruction performance than CDD-RED  if the partially-perturbed example is not towards the successfully adversarial one (namely, p = 1). This is not surprising as DO is optimized using the denoising objective only. However, as p increases (p >= 0.4), we find that CDD-RED indeed outperforms DO; see table below for the metric used in Fig.7(d) between the ground-truth.
>
> | p   | DO      | DS    | CDD-RED (ours) |
> | --- | ------- | ----- | -------------- |
> | 0.0 | **1.24** | 15.27 | 5.07           |
> | 0.2 | **1.03** | 11.64 | 2.78           |
> | 0.4 | 1.93    | 8.90  | **1.73**        |
> | 0.6 | 2.68    | 6.83  | **1.15**        |
>
> Second, even if  DO is closer to the ground-truth than RED at $\text{p} < 0.4$ in terms of the reconstruction error, it does not mean DO is able to recover a more precise adversarial perturbation than CDD-RED. This can be justified from other performance metrics. For example, in Fig.7(b) at p = 0.2, the perturbation estimated by DO achieves a lower attack success rate compared to CDD-RED and the ground truth. Thus, the quality of reverse engineering is worse than ours from the attack performance of the ground truth.
>
> **Q2: The computation cost of RED compared with DS and DO**
>
> **A2:** We measure the computation cost on a single RTX Titan GPU. The inference time for all three methods is similar as they use the same denoiser architecture. For the training cost, the maximum training epoch for each method is set as 300. The average GPU time (in seconds) of one epoch for **DO**, **DS**, and **CDD-RED** is **850**, **1180**, and **2098**, respectively. It is not surprising that CDD-RED is conducted over a more complex RED objective. Yet, the denoiser only needs to be **trained once** to reverse-engineer a wide variety of adversarial perturbations, including those unseen attacks during the training.

---

> ### Author Response · Authors · 2021-11-24
> **Look forward to your post-rebuttal feedback!**
>
> Dear Reviewer T72q,
>
> Thank you very much for taking the time to review our paper. We cherish your comments very much. In our earlier posted response, we have conducted new experiments and added additional clarifications to alleviate your concerns about our original submission. Please refer to our response listed at [Part I](https://openreview.net/forum?id=gpp7cf0xdfN&noteId=c3a7i2SzdVn), [Part II](https://openreview.net/forum?id=gpp7cf0xdfN&noteId=9n-MJvQlbKa), [Part III](https://openreview.net/forum?id=gpp7cf0xdfN&noteId=N4i1YNow7Xl), and [Part IV](https://openreview.net/forum?id=gpp7cf0xdfN&noteId=ii-zg_4rPs) respectively. Please also feel free to find a summary of our response to all reviewers at [Summary Link](https://openreview.net/forum?id=gpp7cf0xdfN&noteId=XS_FHB96yJr) and highlighted general response at [General Response Link](https://openreview.net/forum?id=gpp7cf0xdfN&noteId=X4Ly-ab1yIO).
>
> We hope that you can find our effortful response convincing. If you have additional comments, please feel free to let us know. We will try our best to address them.

---

> > ### Comment · Reviewer_T72q · 2021-11-29
> > **Thank you for a detailed rebuttal**
> >
> > Seconding reviewer e8YB -- thank you for the impressively detailed responses to each of my concerns. I am increasing the score by a point. Good luck!

---

> > > ### Author Response · Authors · 2021-11-29
> > > **Thank you!**
> > >
> > > Dear Reviewer T72q,
> > >
> > > It is our great pleasure to learn that our response has addressed your previous concerns.
> > >
> > > Thank you very much for raising the score.
> > >
> > > Authors,

---

> ### Author Response · Authors · 2021-11-29
> **Post-rebuttal feedback (less than 24 hours)**
>
> Dear Reviewer T72q,
>
> Thank you very much for spending time reviewing our paper. Since the discussion will end very soon, we sincerely hope that you have found time to check our detailed response to your previous questions/comments. If you have any further questions, please feel free to let us know. We will try our best to reply to you before the discussion deadline.
>
> Thank you very much,
>
> Authors

---

### Author Response · Authors · 2021-11-21
**General Response on Applications of RED**

Thank you very much for the very insightful comments. Inspired by these comments, we have made a significant effort to enrich our experiments. In particular, we delved into the potential applications of RED (Reviewer [T72q](https://openreview.net/forum?id=gpp7cf0xdfN&noteId=1hnNnNJVId2) and Reviewer [e8YB](https://openreview.net/forum?id=gpp7cf0xdfN&noteId=FY5YRiP3eZY)).

1) Application 1: We conducted a new experiment to demonstrate how one can leverage the attribution outputs of RED to accomplish the adversarial detection task ([Figure](https://ibb.co/JQ1NWJ2)).

2) Application 2: We conducted a new experiment to demonstrate how the idea of correlation screening (Fig. 8 in the paper) can be leveraged to infer the unseen attack information ([Figure](https://ibb.co/zRtNjgY)).

3) Application 3: We conducted a new experiment to demonstrate the performance of RED approaches applied to certified defense ([Figure](https://ibb.co/cy5yYCL)).

Despite the above-suggested applications, we would like to kindly remark that **adversarial defense is not the purpose of RED**. Instead, we focus on recovering attack perturbation details from adversarial examples. Meanwhile, it is highly non-trivial to build the first formal RED pipeline, including RED formulation, evaluation metric design, RED methodology, and experimentation on the feasibility test of RED. We hope that this can make a considerable impact on the adversarial learning community, as recognized by Reviewer [Ah1k](https://openreview.net/forum?id=gpp7cf0xdfN&noteId=yrd5P4oQB9): "The paper is novel and could have a considerable impact."

---

### Author Response · Authors · 2021-11-21
**Summary of response to all reviewers**

Dear Reviewers, ACs and PCs:

We are glad to receive valuable and constructive comments from all the reviewers. We have made a substantial effort to clarify reviewers' doubts and enrich our experiments in the rebuttal phase. Below is a summary of our responses:

#### Reviewer [T72q](https://openreview.net/forum?id=gpp7cf0xdfN&noteId=1hnNnNJVId2):

1) We provided the computation cost of RED approaches.
2) We provided new experiments on the global setting with a mixture of adversarial, clean, and randomly perturbed data.
3) We conducted new experiments to compare with performing PGD back to the true class.
4) We further clarified the motivation of RED.
5) We provided the potential applications of the extracted perturbations, including building CDD-RED based detector and inferring the attack identity using the reverse-engineered adversarial perturbations.
6) We clarified the results of Fig. 7(d) and Table 2.
7) We clarified the rationale behind RED against Lp attacks and its usefulness.

#### Reviewer [Ah1k](https://openreview.net/forum?id=gpp7cf0xdfN&noteId=yrd5P4oQB9):

1) We provided new experiments on adversarial attacks generated on the CIFAR-10 dataset.
2) We clarified our adaptive attack setting used in the paper to show that an attacker is aware of RED.
3) We provided new experiments to show the RED performance of Wasserstein-type attacks.
4) We fixed the presentation typos.



#### Reviewer [e8YB](https://openreview.net/forum?id=gpp7cf0xdfN&noteId=FY5YRiP3eZY):

1) We further clarified the focus of our paper and the applications of RED, including recovery of adversary’s saliency region and recovery of attack type correlation in the paper.
2) We conducted new experiments to show more applications of RED, including leveraging the outputs of RED to help the design of adversarial detection and attack identity inference.
3) We re-emphasized the contributions of our paper.
4) We conducted new experiments to show the application of RED to provable defense.
5) We explained Fig.8 in the paper to reveal the differences in the distribution of the REDs generated by different attacks.
6) We provided new experiments to compare with the baselines on REDs when facing smoothed classifiers.
7) We provided new experiments on adversarial attacks generated on the CIFAR-10 dataset to show the generalization to smaller datasets.
8) We conducted new experiments to show consistent RED performance against PGD attacks with different hyperparameters settings.

We hope all reviewers find our endeavor in addressing the previous comments and improving our submission.

---

### Public Comment · ~Hossein_Souri1 · 2022-01-31
**A Similar Work**

Thank you for your interesting paper. The key idea of this work is very similar to our recently published paper [1]. Interestingly, we proposed the idea of Reverse Engineering of Deceptions via Residual Learning which is almost the core idea of this work. Please consider mentioning the relationship with our work in your paper, and adding it to the references.

[1] [https://arxiv.org/abs/2110.06802](https://arxiv.org/abs/2110.06802)

---

> ### Public Comment · ~Yuguang_Yao1 · 2022-02-03
> **Response to "A Similar Work"**
>
> Thank you very much for pointing out the related work that enjoys the similar spirit of RED to ours. However, our work is different from the referred one in the following dimensions: We leveraged data augmentation to boost RED performance and made detailed evaluations on RED against unseen attacks from the perspectives of logit alignment, attribution alignment, and reconstruction error. Additionally, we formed the RED pipeline in a full training/testing/application stack. Note that several RED applications in adversarial defense, detection, and inference on attack types are covered. Furthermore, note that the referred paper was published on arXiv after the ICLR22 submission date. Thus, we did not pay much attention to this relevant work. We will cite it in our final version.
>
> Lastly, we are glad to see other work showing that RED is an interesting and important problem.
>
> Thanks for your comment.

---

### Decision · Program_Chairs · 2022-01-20

**Decision:**

Accept (Poster)

**Comment:**

The manuscrupt studies an unexplored problem: How to reverse-engineer adversarial perturbations from an adversarial image? This leads to a new adversarial learning paradigm—Reverse Engineering of Deceptions (RED). The authors formalize the RED problem and identify a set of principles crucial to the RED approach design. By integrating these RED principles with image denoising, they propose a new Class-Discriminative Denoising based RED framework, termed CDD-RED.
The reviewers recognize that this topic is important and a promising research direction.
The reviewers are also satisfied with the respones from the authors.
In summary, this paper is recommended to be accepted as it is well-formulated, easy to follow, and has some merits, despite that it needs to be evaluated further.